# Cancer-associated snaR-A noncoding RNA interacts with core splicing machinery and disrupts processing of mRNA subpopulations

Sihang Zhou [1], Simon Lizarazo[2], Sandip Chorghade[3], Leela Mouli[4], Ruiying Cheng[1], Rajendra K C [5], Auinash Kalsotra [3,6,7,8] & Kevin Van Bortle [1,6] ✉

Expansion of RNA polymerase III (Pol III) activity in cancer can activate the transcription of typically silent small RNA genes, including snaR-A (small NF90-associated RNA isoform A), a hominid-specific noncoding RNA that promotes cell proliferation through unclear mechanisms. Here, we show that snaR-A interacts with mRNA splicing factors, including the U2 small nuclear ribonucleoprotein (snRNP) subunit SF3B2, and localizes near subnuclear foci enriched in splicing machinery. Overexpression of snaR-A increases intron retention, a hallmark of inefficient splicing, whereas its depletion enhances splicing of mRNAs characterized by high U2 snRNP occupancy and nuclear speckle proximity. These improvements in splicing coincide with reduced cell proliferation, consistent with tumor-level patterns linking snaR-A to growth in primary cancers. Together, these findings identify snaR-A as a molecular antagonist of splicing and potential disease driver in cancer. We propose that snaR-A-related splicing perturbation may phenocopy splicing defects attributed to U2 snRNP mutations in cancer, eliciting an alternative, non-mutational mechanism of splicing dysregulation during tumorigenesis.

The Pol III transcriptome includes multiple classes of small, highly structured noncoding RNA species involved in diverse cellular processes. For example, tRNA and 5S, the smallest ribosomal RNA (rRNA), play integral roles in translation and protein accumulation. Other Pol III-transcribed genes encode ncRNAs involved in splicing (U6, U6atac), transcription regulation (7SK), RNA processing and stability (H1, RMRP, Y RNA), as well as autophagy (Vault). The signal recognition particle RNA (7SL), on the other hand, targets secretory proteins for translation and appropriate folding at the endoplasmic reticulum. In this way, Pol III transcription produces a multitude of RNAs with core housekeeping functions[1–3].

In humans, retrotransposition of the highly expressed 7SL RNA has established hundreds of 7SL pseudogenes, which are thought to be transcriptionally and functionally incompetent[4]. Through subsequent duplications and sequence divergence, Alus—the most abundant class

[1]Department of Cell and Developmental Biology, University of Illinois Urbana-Champaign, Urbana, IL, USA. [2]Department of Molecular and Integrative Physiology, University of Illinois Urbana-Champaign, Urbana, IL, USA. [3]Department of Biochemistry, University of Illinois Urbana-Champaign, Urbana, IL, USA. [4]School of Molecular and Cellular Biology, University of Illinois Urbana-Champaign, Urbana, IL, USA. [5]Center for Biophysics and Quantitative Biology, University of Illinois Urbana-Champaign, Urbana, IL, USA. [6]Cancer Center at Illinois, University of Illinois Urbana-Champaign, Urbana, IL, USA. [7]Carl R. Woese Institute for Genomic Biology, University of Illinois Urbana-Champaign, Urbana, IL, USA. [8]Chan Zuckerberg Biohub Chicago, LLC, Chicago, IL, USA. ✉e-mail: kvbortle@illinois.edu

of Short Interspersed Nuclear Elements (SINEs)—evolved and further proliferated[5]. Two hominid-specific Pol III-transcribed genes, BC200 and snaR, are evolutionarily derived from Alu sequences. The expression of BC200 and snaR is restricted to specific contexts, with BC200 expressed in neurons and snaR in testes[6,7]. Despite prior evidence that both ncRNAs also re-emerge in human tumors, functional characterization of BC200 and snaR in tissue- and cancer-specific contexts remains limited.

snaR, an acronym for small NF90-associated RNA, was first reported in 2007 as an RNA species upregulated in EBV-infected B cells

and enriched in biochemical pulldown experiments targeting an RNA-binding protein, ILF3[8,9]. Multiple isoforms for snaR ranging from snaR-A to snaR-I are encoded in multiple clusters of repetitive sequence elements, with snaR-A the dominant isoform expressed in testes and cancers (Fig. 1a, Supplementary Fig. 1)[7,10]. Recent reports indicate that snaR-A promotes cell proliferation, migration, and invasion phenotypes in multiple contexts[11–18]. Thus, deconstructing the molecular function and consequence of snaR-A emergence remains critical for understanding how snaR-A may contribute to cancer initiation and progression.

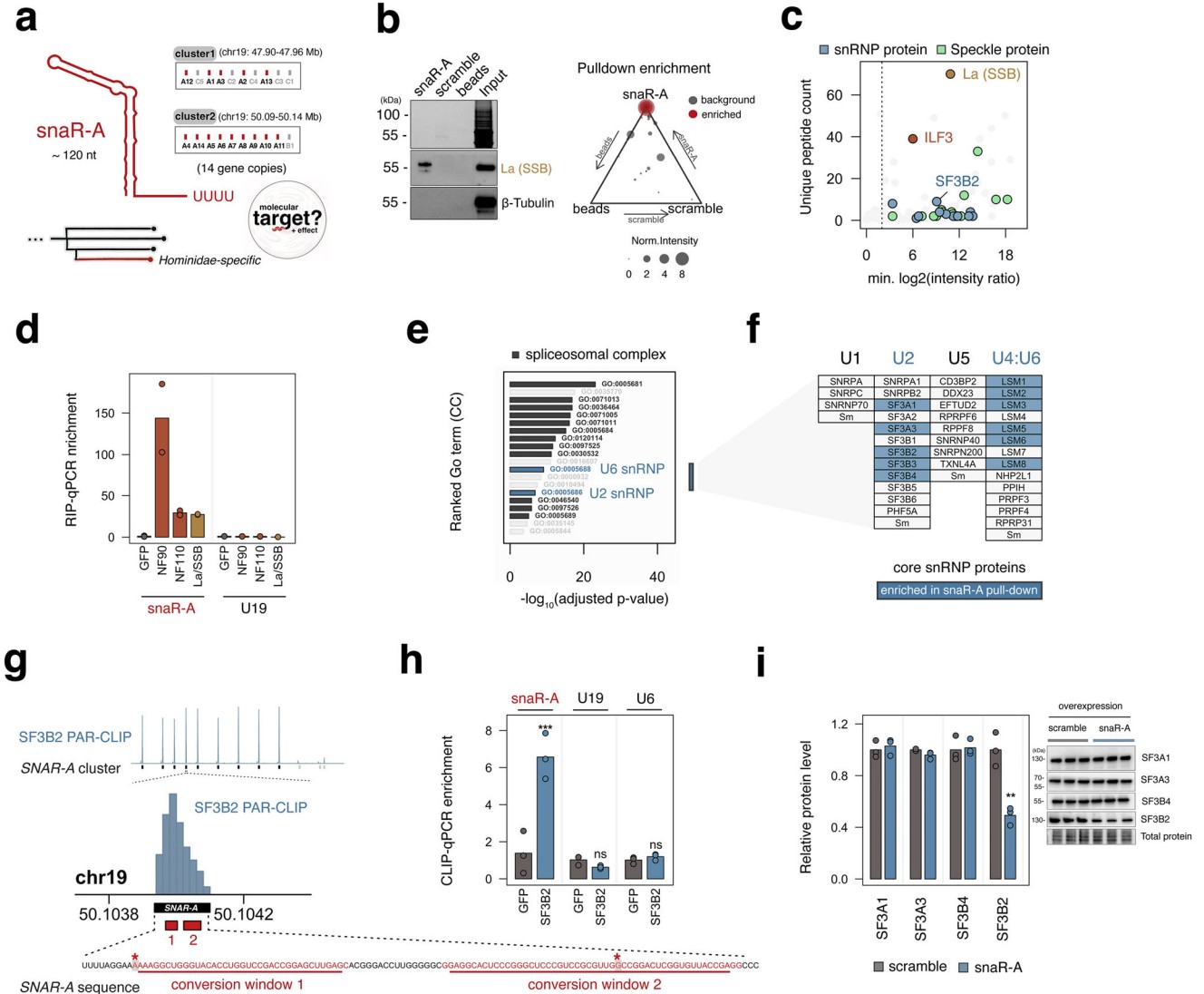

**Fig. 1 | snaR-A ncRNA interacts with mRNA splicing machinery, including U2 snRNP protein SF3B2. a** Illustrative overview of established snaR-A features and the present study aim of delineating the molecular consequence of snaR-A expression in cells. **b** Immunoblot showing total protein, RNA chaperone protein La, and beta-tubulin in snaR-A, scramble, or empty beads pull-down experiments. Triangle plot depicts normalized intensity of proteins enriched for snaR-A binding (red) and non-enriched proteins (gray). **c** Minimum log2(intensity ratio) and total unique peptides counts in snaR-A pull-down mass spec. ILF3, La, and splicing-associated proteins are highlighted. **d** RNA Immunoprecipitation (RIP)-qPCR quantification of target RNA enrichment (snaR-A, U19) in Flag-NF90, Flag-NF110, and Flag-La (SSB) RIP experiments compared to Flag-GFP, relative to the enrichment of control small ncRNA, Z30. **e** Gene Ontology Cellular Component (GO:CC) enrichment analysis for significantly enriched proteins in snaR-A RNA-pull-down experiments. Gene ontology enrichment analysis was performed using one-sided

hypergeometric test. *p*-values were adjusted for multiple testing using the Benjamini-Hochberg method. **f** Visualization of core snRNP proteins for U1, U2, U5, and U4/U6, with all identified snaR-A interacting proteins highlighted in blue. **g** Visualization of SF3B2 PAR-CLIP read pileup across snaR-A gene clusters, and inspection of sequence conversion windows indicative of direct RNA-protein cross-linking sites. (*asterisk denotes coordinate with maximum enrichment).
**h** Crosslinking immunoprecipitation (CLIP)-qPCR quantification of target RNA enrichment (snaR-A, U19, U6 snRNA) in Flag-SF3B2 CLIP experiments compared to Flag-GFP, relative to the enrichment of control small ncRNA, Z30. **i** Immunoblot and corresponding quantification of U2 snRNP protein levels for SF3A1, SF3B1, SF3B4, and SF3B2 in HEK293T overexpressing snaR-A. For RIP (**d**), biological replicates = 2; for CLIP (**h**) and immunoblot (**i**), biological replicates = 3; two-group comparison analyzed with two-sided unpaired *t* test; *p ≤ 0.05; **p ≤ 0.01; ***p ≤ 0.001; ****p ≤ 0.0001. Source data are provided as a Source Data file.

Here, we investigated snaR-A-centric protein interaction preferences with the expectation that, like other small ncRNAs, snaR-A primarily functions via specific ribonucleoprotein complex(es). Using unbiased biochemical approaches, we unexpectedly uncover widespread interactions between snaR-A and mRNA splicing factors. Our ensuing cellular and functional genomic study provides evidence that snaR-A interactions drive splicing inefficiencies in specific mRNA subpopulations linked with cancer progression. snaR-A-driven perturbations reduce protein abundance for a wide-ranging set of factors, suggesting the emergence of snaR-A disrupts cellular homeostasis via perturbation of splicing. We further show that snaR-A primarily enhances cell proliferation rather than migration, altogether expanding our understanding of the nature of snaR-A activity and the link between Pol III overactivity, cancer initiation, and disease progression.

## Results

### Integrated proteomic and genomic discovery of snaR-A RNA-protein interactions

Many Pol III-derived small ncRNAs were discovered as core RNA components of specific ribonucleoprotein complexes (RNPs), typically central to the established roles of each respective ncRNA[3]. Similarly, the origin story of snaR-A includes its discovery as an RNA molecule enriched in biochemical pulldown experiments for NF90, a specific isoform of the RNA-binding protein, ILF3. Though NF90 interactions function to chaperone and thereby stabilize snaR-A[7,19], whether additional, presently uncharacterized snaR-A- binding factors are more centrally involved in its potential biological role remain largely unexplored. We therefore sought to discover snaR-A RNA-protein interactions de novo, using 5′ biotinylated snaR-A to pull down interacting proteins from THP-1 cell lysates followed by mass spectrometry (MS) (Fig. 1b). In total, our in vitro pull-down assay yielded 375 protein hits with 61 exhibiting significant enrichment in snaR-A pull downs compared to both empty bead and scramble RNA control experiments (Fig. 1b, Supplementary Data 1). Most of the 61 snaR-A-protein interactions identified correspond to a multitude of RNA-binding proteins (RBPs), with La (SSB), an RNA-chaperone protein important for processing and stability of Pol III-transcribed genes, the most confident hit (Fig. 1b, c)[20]. ILF3 (NF90) is also notable among the list of significantly enriched proteins, indicative of snaR-A-NF90 interactions in THP-1 monocytes (Fig. 1c). We confirm snaR-A-La and snaR-A-ILF3 interactions through reciprocal pulldown experiments, in which flag-tagged La and ILF3 isoforms and associated RNA are captured by RNA immuno-precipitation (RIP) (Fig. 1d). Here, snaR-A RNA is enriched in pulldown experiments for either NF90 or NF110 isoforms, suggesting snaR-A-ILF3 interactions are not strictly isoform-dependent (Fig. 1d).

To complement our proteomics result, we further conducted a large-scale survey for snaR-A enrichment in genomic RNA cross-linking and immunoprecipitation (e.g. eCLIP and CLIP-seq) experiments[21]. Using this approach, which integrates RNA-binding patterns for >150 unique RBPs, we find that both La (SSB) and ILF3 eCLIP experiments are among the highest with respect to snaR-A enrichment (Supplementary Fig. 2), further supporting these interactions and the validity of our in vitro binding assay.

### snaR-A interacts with mRNA splicing machinery, including U2 snRNP protein SF3B2

Beyond uncovering interactions with chaperone protein La and reaffirming ILF3-snaR-A binding, our proteomic analysis unexpectedly pointed to enrichment for a multitude of splicing factors (Fig. 1c). Gene ontology (GO) enrichment analysis confirms overrepresentation of numerous spliceosomal complex-related terms and, more specifically, U2 and U6 snRNP proteins (Fig. 1e). Among the core U1, U2, U5, and U4/ U6 snRNP proteins, snaR-A interactions were indeed specific to U2 and U4/U6 components (Fig. 1f). However, revisiting our large-scale eCLIP

survey, we find that experiments for SF3B1, SF3B4, and SF3A3−components of the U2 snRNP complex−are not enriched for snaR-A ncRNA, limiting the likelihood of direct interactions between snaR-A and these particular proteins (Supplementary Fig. 2). In contrast, analysis of PAR-CLIP (Photo Activatable Ribonucleoside-enhanced) data for SF3B2, another U2 snRNP protein, uncovers nucleotide-level evidence of direct snaR-A-SF3B2 interactions (Fig. 1g)[22]. Immunoblot analysis of SF3B2 confirms biotin-snaR-A-dependent pull-down, whereas CLIP-qPCR analysis against flag-tagged SF3B2 reciprocally supports snaR-A-SF3B2 interactions in human embryonic kidney 293 T (HEK293T) cells−in contrast to U6 and U19 snRNAs, which are not enriched in SF3B2 pull-down experiments (Fig. 1h). Moreover, we show that snaR-A overexpression results in subunit-specific depletion of SF3B2, in contrast to U2 snRNP proteins SF3A1, SF3A3, and SF3B4 (Fig. 1i). These data altogether establish an unexpected link between snaR-A and mRNA splicing machinery, specifically U2 snRNP protein SF3B2, prompting us to further investigate the subcellular localization of snaR-A and spatial relationship to nuclear splicing processes.

### snaR-A ncRNA localizes to subnuclear foci associated with mRNA splicing

To visualize snaR-A, we applied the hybridization chain-reaction RNA fluorescence in situ hybridization (HCR-RNA-FISH) method, which amplifies signal through cascade polymerization mechanisms (Fig. 2a)[23]. This approach facilitates the detection of small RNA species for which standard RNA FISH methods (which rely on collections of hybridization probes that target a larger sequence space) are unfeasible. We note that HCR-RNA-FISH detects snaR-A localization near subnuclear foci, supporting a role for snaR-A in nuclear-related processes (Fig. 2b). These observations are further supported by subcellular fractionation small RNA-seq experiments, which capture significant differential enrichment of snaR-A within purified nuclei, rather than cytoplasmic fractions in this context (Supplementary Fig. 3). We note that nuclear snaR-A abundance is particularly sensitive to perturbation of chaperone protein La, suggesting La plays an important role in snaR-A localization or stability (Supplementary Fig. 3).

In metazoans, the subnuclear organization of splicing includes nuclear "speckles"−membrane-less nuclear bodies enriched for splicing factors and higher activity of co-transcriptional splicing[24-27]. We therefore considered whether nuclear snaR-A is positionally related to nuclear speckles by co-staining for SON, a scaffolding component of nuclear speckles[28]. Co-staining reveals that snaR-A RNA spatially surrounds nuclear bodies marked by SON (Fig. 2b), and that the convergence of snaR-A and SON signal is significant in multiple cell types, including HEK293T and A549 lung adenocarcinoma cells (Fig. 2c, Supplementary Fig. 4). snaR-A enrichment adjacent to nuclear speckles is additionally confirmed by orthogonal co-staining of SC35 (SRSF2), an serine/arginine-rich (SR) family protein highly enriched within speckles (Fig. 2c, d)[26]. snaR-A foci are equivalently enriched for the U2 snRNP protein SF3B2, which is similarly observed at the periphery of nuclear speckles and, like snaR-A, at subnuclear foci independent of splicing bodies, consistent with previously reported subnuclear distributions of SF3B1 (Fig. 2c, d)[29]. HCR-RNA-FISH visualization of U6 snRNA also detects subnuclear foci surrounding nuclear speckles, suggesting snaR-A patterns are consistent with that of core splicing proteins and small nuclear RNAs (Fig. 2e). We additionally note the particularly strong co-localization patterns observed between snaR-A and U6 snRNA (Fig. 2c, e). This pattern is consistent with the enrichment of heptameric (U6-bound) Lsm proteins in snaR-A pulldown experiments (Fig. 1f) and suggests that snaR-A may be intricately linked with multiple splicing factors.

### snaR-A genes are spatially proximal to nuclear speckles

Tyramide signal amplification coupled with sequencing (TSA-seq) is a genomic method that measures subnuclear proximity of DNA

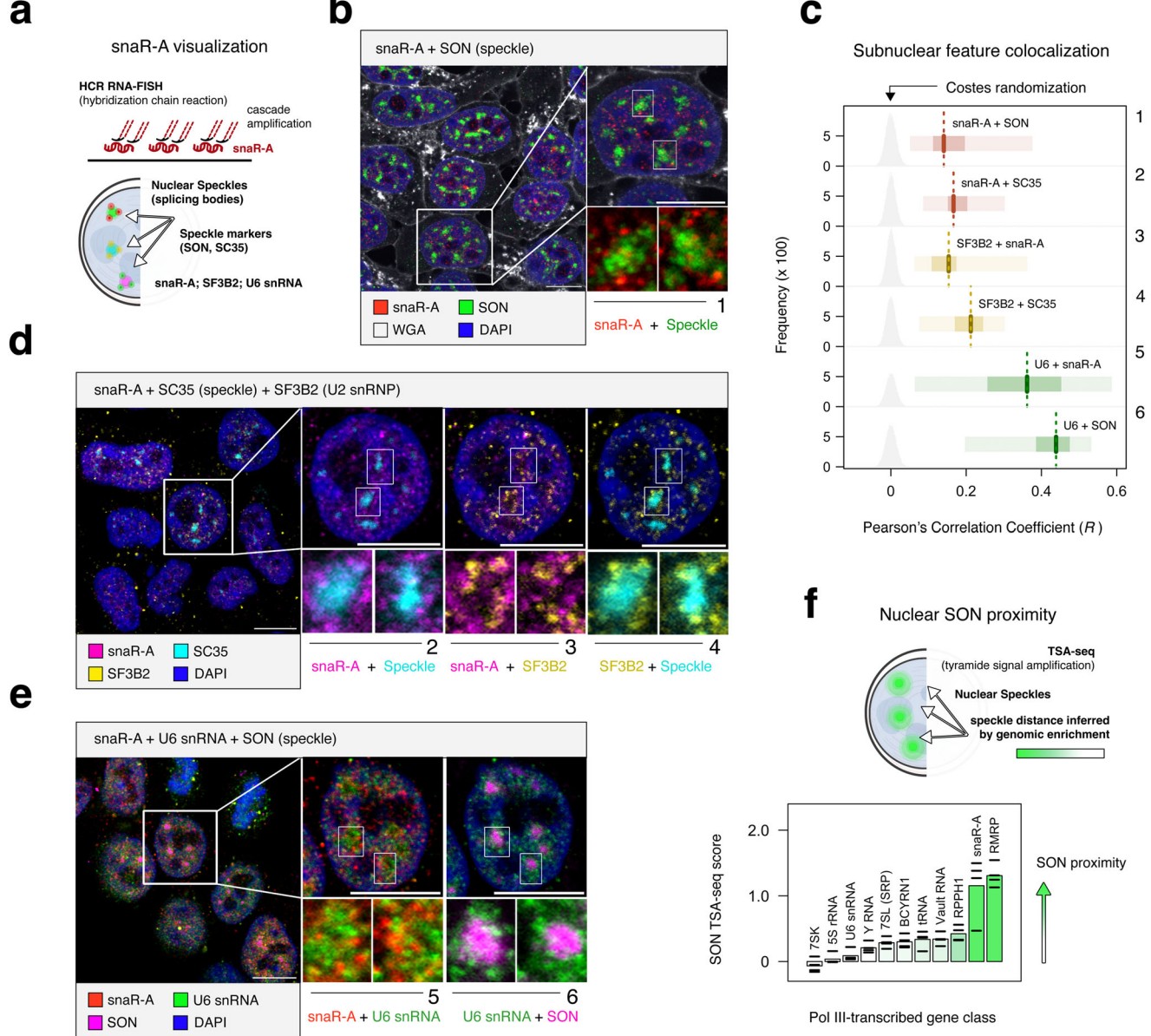

**Fig. 2 | snaR-A localizes to subnuclear foci that are associated with splicing bodies (speckles) and related factors. a** Illustrative guide to snaR-A hybridization chain reaction-fluorescence in situ hybridization (HCR-RNA-FISH) experiments and co-staining for nuclear speckle markers (SON, SC35), SF3B2 (U2 snRNP), and U6 snRNA. **b** HCR-RNA-FISH detection of snaR-A, co-stained for SON (nuclear speckles), wheat germ agglutinin (WGA; membrane), and DAPI. Scale bars, 10 μm (n = 2 biological replicates). **c** Quantitative analysis of in situ feature co-localization. Observed correlation distributions are compared against Costes randomization-based null hypotheses. Individual plots correspond to co-localization for snaR-A,

SON, SC35, SF3B2, and U6 snRNA presented in (**b**, **d**, **e**). Box plots show the median (center), the 25th and 75th percentiles. Light shade present minimum and maximum values. (n = 30 cells). **d** HCR-RNA-FISH detection of snaR-A, co-stained for SC35 (SRSF2, nuclear speckles), SF3B2, and DAPI. Scale bars, 10 μm (n = 2 biological replicates). **e** HCR-RNA-FISH detection of snaR-A and U6 snRNAs, co-stained for SON (nuclear speckles). Scale bars, 10 μm (n = 2 biological replicates). **f** Analysis of snaR-A gene proximity to nuclear speckles on the basis of SON TSA-seq, a genomic method that infers sequence proximity following tyramide signal amplification directed by anti-SON antibodies. Source data are provided as a Source Data file.

sequence to specific nuclear bodies and subcompartments. For example, SON TSA-seq experiments map genomic distance to nuclear speckles by targeting TSA reactions via antibodies against the speckle scaffold protein, SON[30]. Thus, a high SON TSA-seq enrichment for a given gene locus indicates close association with nuclear speckles. Analysis of the snaR-A genes on chromosome 19 finds that snaR-A are characterized by notably strong TSA-seq enrichment scores, particularly in comparison to all other classes of Pol III-transcribed genes (Fig. 2f). These data suggest that, complementary to the broader patterns of snaR-A spatial proximity to splicing bodies, nascent snaR-A may be uniquely and optimally positioned to interact with and potentially influence nuclear splicing processes. We, therefore, next

investigated the global consequences of snaR-A ncRNA abundance on mRNA splicing events.

**Increased snaR-A levels are linked with elevated intron retention (IR), a hallmark of inefficient splicing**
To determine the consequences of snaR-A expression on splicing, we overexpressed snaR-A in either HEK293T or A549 cells and quantified changes in mRNA processing through ultra-deep RNA-seq experiments (Fig. 3a). Given our finding that snaR-A overexpression reduces SF3B2 abundance (Fig. 1i), we additionally overexpressed SF3B2 to understand the independent effects of snaR-A and SF3B2 levels on splicing outcomes (Fig. 3b, Supplementary Fig. 5). We prioritized downstream

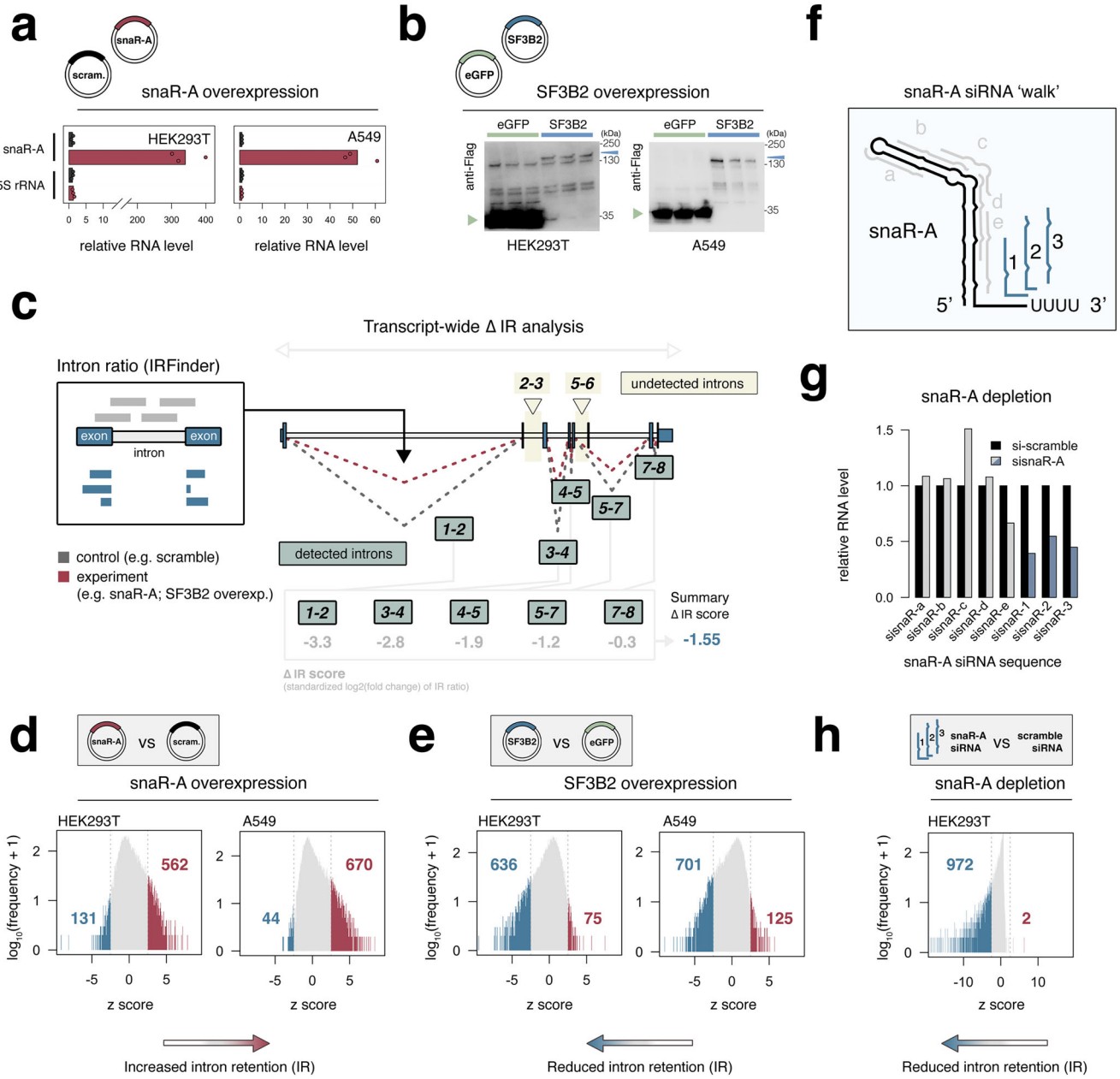

**Fig. 3 | snaR-A expression is associated with higher levels of intron retention (IR), consistent with perturbations of mRNA splicing efficiency. a** RT-qPCR analysis of snaR-A RNA and 5S rRNA levels in HEK293T or A549 cells transfected with snaR-A (red) or scramble RNA (black) overexpression constructs. (*n* = 3 biological replicates). **b** Immunoblot analysis of SF3B2 and eGFP protein abundance in HEK293T or A549 cells transfected with flag-tagged SF3B2 (blue) or flag- tagged eGFP (green) overexpression constructs (*n* = 3 biological replicates). **c** Overview of transcript-level intron retention (IR) analysis. IRFinder intron ratios are compared between experiment (red) and control treatments (gray). Delta IR values computed as standardized log2(fold change) are summarized for (*n*) intron events and compared against a randomization-based null distribution of (*n*) introns. **d**, **e** Distributions of transcript-level delta IR z-scores in HEK293T or A549 cells following snaR-A (**d**) or SF3B2 overexpression (**e**). **f** Illustrative overview of siRNAs and related target regions tested against snaR-A. **g** RT-qPCR analysis of snaR-A knockdown results using siRNAs illustrated in panel f. siRNA #1, #2, and #3 successfully reduced snaR-A levels by >= 50%. **h** Distribution of transcript-level delta IR z-scores in HEK293T cells following snaR-A depletion using siRNA #1, #2, or #3. Source data are provided as a Source Data file.

---

analysis of intron retention (IR)—a feature that reflects the ratio of unspliced-versus-spliced transcripts—because IR patterns most accurately reflect all intronic splicing events (i.e. splicing efficiency) rather than isoform-specific exon usage signatures[31]. We additionally computed a transcript-centric dynamic IR score that compares the distribution of intron ratios, determined via IRFinder following snaR-A or SF3B2 overexpression across all annotated introns for a given gene, allowing for broad comparisons of transcript-level splicing efficiencies (Fig. 3c)[32]. Using this framework, we show that snaR-A overexpression produces a shift towards higher IR levels (Fig. 3d), whereas SF3B2

overexpression produces a contrasting pattern of IR reduction in both HEK293T and A549 cells (Fig. 3e). These results altogether suggest a model in which snaR-A ncRNA disrupts efficient splicing through interactions with SF3B2 and potentially other splicing factors.

### snaR-A depletion reduces IR within mRNAs marked by U2 snRNP residency and nuclear speckle proximity

In complement with our overexpression experiments, we also depleted endogenous snaR-A using siRNAs designed against 3'-proximal sequences, the only interval capable of snaR-A reduction compared to

siRNAs targeting other regions within the small stem-loop structure of snaR-A (Fig. 3f, g). Ensuing RNA-seq and transcript-centric analyses, which here integrates IRFinder statistics derived for each independent siRNA, reveals a broad reduction of IR levels following snaR-A depletion, similar in nature to SF3B2 overexpression (Fig. 3h). To gain insight into the particular mRNA features associated with significant IR reduction, we applied a logistic regression analysis that considers a multitude of intrinsic and extrinsic features, including sequence composition, intron length, RBP-binding patterns, and speckle proximity. Whereas the presence of U2 snRNP proteins (SF3B4 and SF3A3) and nuclear speckle proximity are among the strongest positive predictors of IR reduction following snaR-A depletion, we find that intron length is the strongest negative predictor (Fig. 4a).

The recovery of U2 snRNP binding and nuclear speckle proximity as predictors of transcript-level IR reduction, which scale proportionally with splicing improvement models (Fig. 4b, c), adds further supporting evidence that snaR-A likely perturbs mRNA processing through interactions with specific splicing factors, either at or in proximity to nuclear splicing bodies. We therefore sought to validate specific examples of splicing improvement for transcripts marked by significant U2 residency and nuclear speckle proximity (Fig. 4c). To this end, we also complemented our transcript-centric IR findings with results from rMATS, a tool that quantifies IR and other forms of alternative splicing[33], to independently identify significant differential splicing events. Importantly, transcript-level and intron-centered events called by rMATS overlap significantly, and together feature overrepresentation of transcripts encoding NuRD (chromatin remodeler and deacetylase) complex subunits and factors involved in autophagy (Fig. 4d, e). We show that many such differential IR events are supported by exon-junction PCR experiments for specific transcripts, including those encoding MTA1 (NuRD regulatory subunit), MCRIP2 (putative stress granule protein), OGFR (opioid growth factor receptor), AUP1 (regulator of lipid metabolism and autophagy), and a host of other factors (Fig. 4f, g, Supplementary Fig. 7)[34–37]. In many cases, splicing improvements (i.e. reduced IR following snaR-A depletion) lead to detectable increases in protein abundance (Fig. 4g), suggesting snaR-A-related splicing defects modulate both the production of mature mRNA and downstream protein abundances. Among these changes, the splicing improvements and upregulation of OGFR protein levels may be particularly noteworthy in the context of snaR-A and cancer, given that OGFR is a well-established inhibitor of cell proliferation and tumorigenesis[38]. We therefore directed our attention to the link between snaR-A expression and splicing with cancer-related phenotypes.

## snaR-A depletion reduces proliferation but enhances cell migration in HEK293T cells

To date, depletion of snaR-A has been shown to reduce cell proliferation in multiple contexts, including THP-1 monocytes and both MDA-MB-231 and SK-BR3 breast cancer cell lines[12,13,16]. snaR-A is also linked with enhanced cell migration, potentially through a non-canonical miRNA-related pathway[17]. We therefore investigated the effect of snaR-A depletion on proliferation and migration phenotypes in HEK293T, relevant to our observed changes in splicing. As shown for other contexts, snaR-A depletion significantly reduces HEK293T proliferation, reflected by overall reductions in cell count, replicating cell count (EdU+ cells), and proliferation index measurements (replicating EdU+ fraction; $p = 0.0132$, 0.0018, and 0.0111 respectively, two-way ANOVA, Fig. 5a). These findings are in line with previous reports and together indicate that snaR-A may broadly enhance cell proliferation. In contrast, transwell migration assays reveal significantly increased cell motility in response to snaR-A depletion, suggesting snaR-A does not promote cell migration phenotypes in this context (Fig. 5b). Though we speculate that increased migration may reflect the impact of a cell proliferation-migration trade-off in this context, these results

help to distinguish snaR-A as an acute driver of proliferation, rather than migration, in this particular context.

## snaR-A gene-activity signatures are linked with cell proliferation, not migration, in primary tumors

Beyond previous reports of its expression in testes, a broad survey of snaR-A across cancer subtypes remains currently lacking, in part due to its small, highly structured nature[10] and absence in traditional RNA-seq experiments, its omission from certain gene annotation sets, as well as its repetitive multi-copy gene origins[7,9]. To circumvent these limitations, which currently preclude a large-scale RNA-based survey of snaR-A abundance in cancer, we devised a chromatin-based statistical framework to determine the likely expression state (i.e., active vs. inactive) of snaR-A genes on the basis of gene accessibility, which is known to correlate with both RNA polymerase III occupancy and transcription[39]. Using this approach, we annotated snaR-A and other Pol III-transcribed gene states across 328 primary tumors[40]. We find that snaR-A is most frequently active in Testicular Germ Cell Tumors (TGCT; Fig. 5c), an observation that is consistent with high snaR-A abundance reported in human testes[7]. The snaR-A gene is otherwise commonly active across > 50% of Colon adenocarcinoma (COAD, 73%), Head and Neck squamous cell carcinoma (HNSC, 69%), and Esophageal carcinoma (ESCA, 51%), and most frequently silent in Glioblastoma multiforme (GBM, 3%), Mesothelioma (MESO, 6%), and Thyroid carcinoma (THCA, 7%; Fig. 5c).

Given that snaR-A depletion reduced cell proliferation rather than migration in our particular context, we sought to broadly characterize the relationship between snaR-A gene activity, proliferation, and migration across primary tumors. We specifically considered key markers that are either upregulated during epithelial-mesenchymal transition (EMT)—an indirect proxy for tumor migration and invasion phenotypes—or other gene markers commonly shared by cell proliferation programs[41,42]. In contrast to EMT markers, which lack any association with snaR-A hallmarks in tumors, we find that snaR-A positively correlates with cell proliferation markers across these contexts, with a gene distribution mean of $r = 0.237$ (Spearman's rho; $p < 1e$ −6, permutation test, Fig. 5d). These findings are consistent with cell-based proliferation assays and lend additional support to a model in which snaR-A primarily promotes cell expansion rather than migration phenotypes in cancer.

## snaR-A gene activity signatures and splicing defects are enriched in cancer-linked mRNA subpopulations

Finally, to explore the potential impact of snaR-A on disease progression, we applied a Cox proportional hazard regression for snaR-A gene state across tumors, restricting to the cancer subtypes with variability in predicted gene on or off status[43]. Briefly, cancers with predicted uniform ON and uniform OFF states, defined using the median statistic across all 14 snaR-A genes, were excluded from further analysis to mitigate cancer subtype biases. Cancer subtypes with variable expression were subsequently restricted to tumor samples with confident ON and confident OFF states, which were thereafter used as input for survival analysis. Using this approach, we find that snaR-A is indeed a negative predictive factor in cancer, with a median hazard ratio of 1.52 (Fig. 5e, $n = 187$, HRs range between 1.19 to 1.71). Most notably, snaR-A genes are characterized by a comparatively high median Wald statistic in comparison to Pol III-transcribed gene types and other small ncRNA genes (Fig. 5f, $z = 2.46$). These findings indicate a link between individual, tumor-specific snaR-A expression markers and unfavorable outcomes in patients, once again consistent with a model in which snaR-A promotes disease progression.

The overarching pattern of snaR-A emergence, splicing disruption, proliferation phenotypes and negative disease signatures leads us to ultimately ask whether the mRNA subpopulations with snaR-A-related splicing defects themselves are linked in any way with cancer

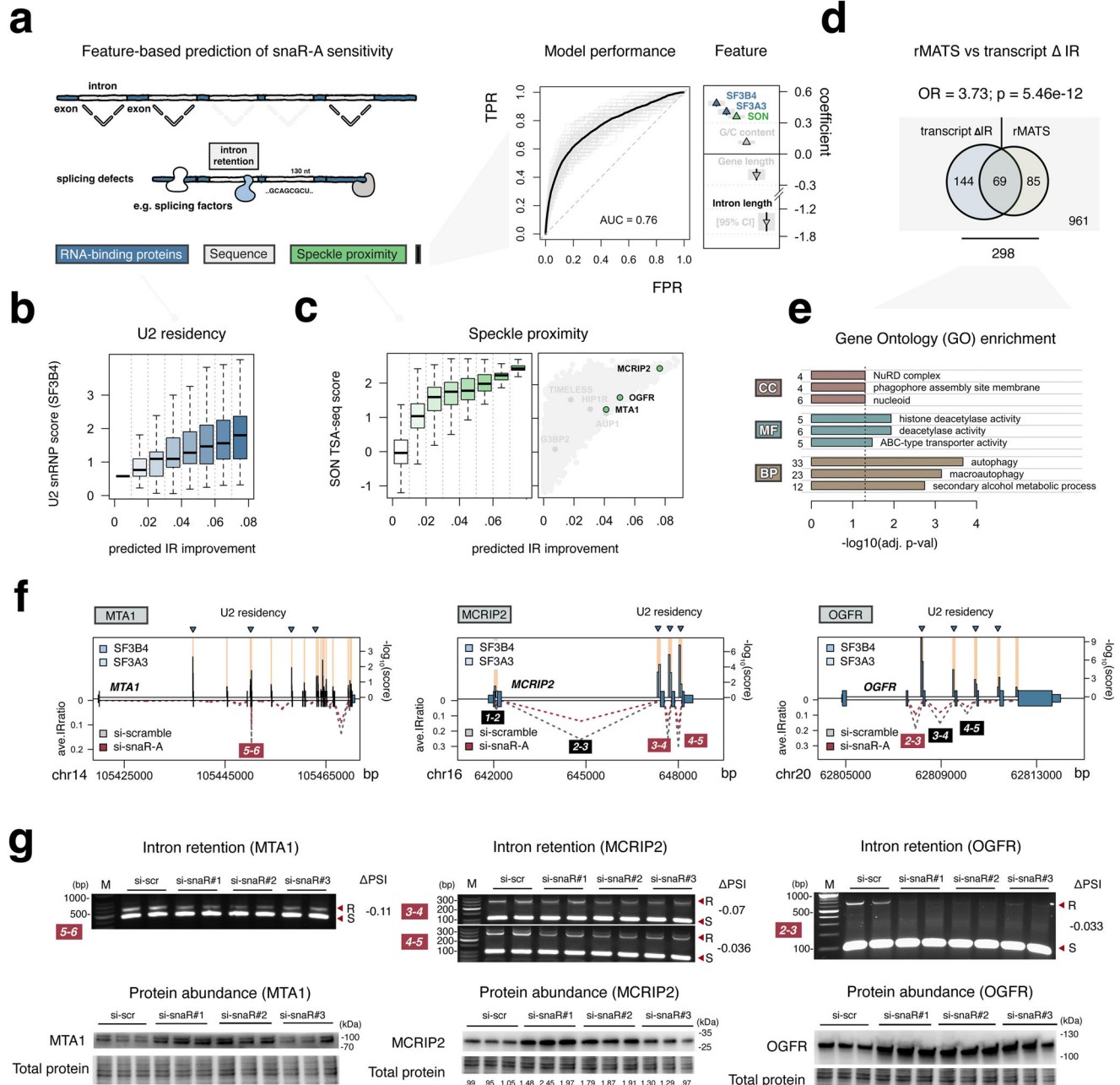

**Fig. 4 | snaR-A depletion improves splicing hallmarks in mRNA subpopulations associated with U2 residency and nuclear speckle proximity. a** Illustrative overview of intrinsic and extrinsic features considered for logistic regression analysis (left) and model performance of snaR-A-sensitive classification (right). Regression coefficients for notable features, including U2-binding (SF3B4, SF3A3 eCLIP), speckle proximity (SON TSA-seq), and intron length are highlighted. Data are presented as median values with 95% confidence intervals. (bootstrap $n = 100$). **b, c** U2 snRNP (**b**) and SON TSA-seq scores (**c**) binned by predicted intron retention (IR) improvement following snaR-A depletion. Box plots show the median (center), the 25th and 75th percentiles (bounds of the box), and whiskers extending to values within 1.5× the interquartile range. ($n = 6266$ [0,0.01], $n = 2352$ [0.01,0.02], $n = 1044$ [0.02,0.03], $n = 498$ [0.03,0.04], $n = 214$ [0.04,0.05], $n = 142$ [0.05,0.06], $n = 59$ [0.06,0.07], $n = 48$ [ > 0.07]). **d** Overlap of transcript-level IR analysis with individual

IR events called by rMATS ($p = 5.46e{-}12$; Two-sided Fisher's Exact Test). **e** Gene ontology (GO) enrichment analysis on transcripts identified in (**d**). Gene ontology enrichment analysis was performed using one-sided hypergeometric test. $p$-values were adjusted for multiple testing using the Benjamini-Hochberg method. **f** Transcript structure, U2-binding, and intron retention levels for snaR-A sensitive transcripts MTA1 (a NuRD complex subunit), MCRIP2 (stress granule protein), and OGFR (opioid growth factor receptor; tumor suppressor gene). **g** Exon–exon junction PCR analysis for intronic transcript features highlighted in (**f**), and the corresponding changes in splicing following snaR-A depletion (PSI = percent spliced in; top); corresponding immunoblot analysis of protein abundance for MTA1, MCRIP2, and OGFR shown at bottom. Source data are provided as a Source Data file.

progression. We specifically explored univariate Cox proportional hazard models reported by Smith and Sheltzer, which quantify the relationship between mRNA mutations (including splice site and other non-synonymous mutations) and patient outcomes reported by The Cancer Genome Atlas (TCGA)[40]. In brief, we determined the frequency

of genes with either negative or positive mutation-based prognostic power and compared these frequencies for specific mRNA subpopulations to randomized null distributions. Our results demonstrate significant overrepresentation of both negative and positive prognostic genes within the mRNA subpopulations characterized by snaR-

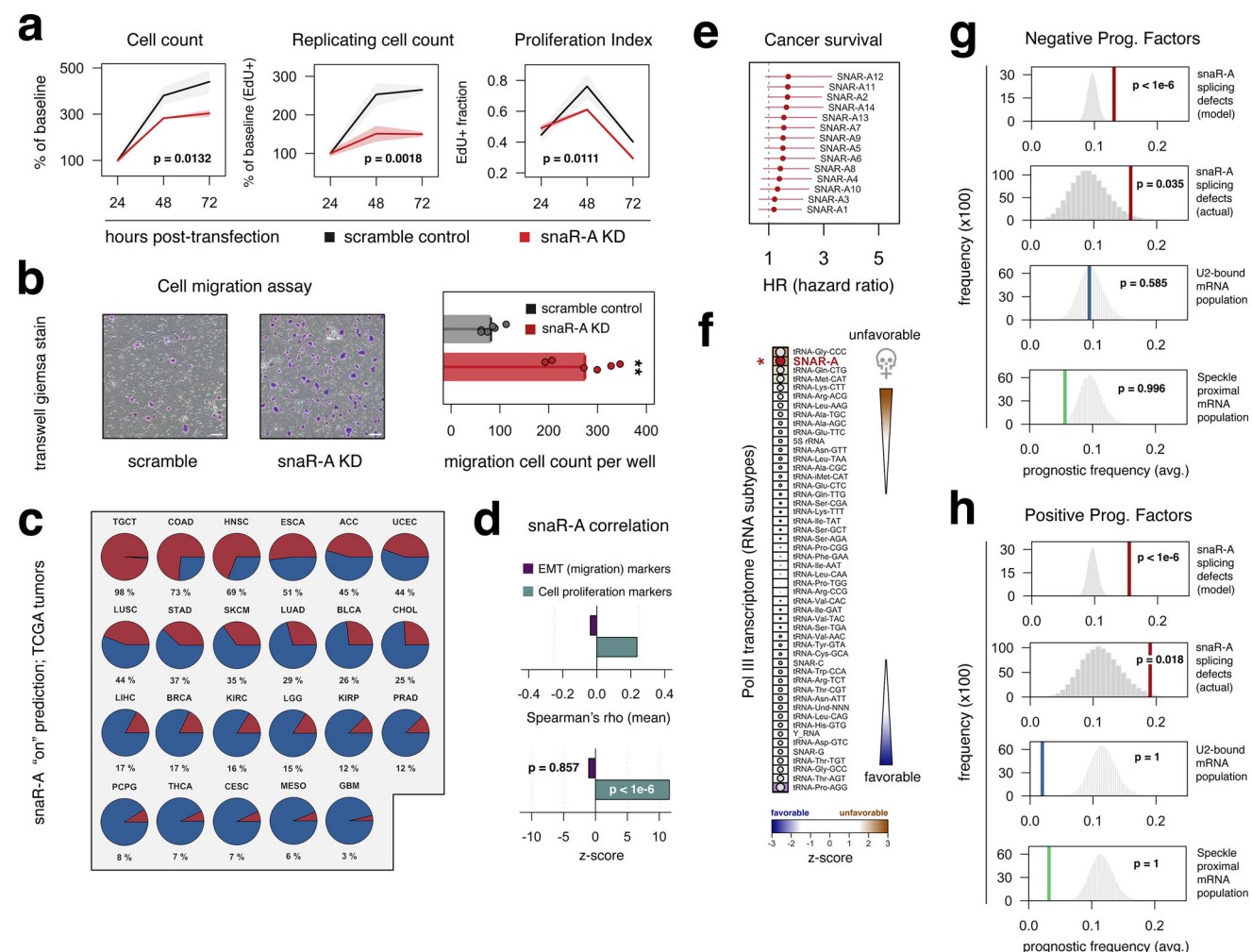

**Fig. 5 | snaR-A promotes cell proliferation and is a negative prognostic factor in cancer. a** Cell proliferation and EdU (5′-ethynyl-2′-deoxyuridine) incorporation assay following snaR-A depletion (red), compared to scramble control (black). Cell count ($p = 0.0132$), EdU+ count ($p = 0.0018$), and EdU+ fraction (proliferation index determined as percentage of EdU+ nuclei visualized with DAPI, $p = 0.0111$, Two-way ANOVA) were measured at 24-, 48-, and 72-h post-transfection. Data are shown as mean values with shaded areas representing the minimum and maximum observed values. ($n = 2$ biological replicates). (**b**) Transwell migration assay and ensuing cell count with giemsa stain following snaR-A depletion (red), compared to scramble control (black). **$p < 0.001$, two-sided unpaired t test. Scale bars, 200 μm ($n = 6$ biological replicates). **c** High-to-low cancer-specific frequencies of predicted *SNAR-A* gene ON (red) and OFF (blue) state across 23 human tumor subtypes. See methods for corresponding names of abbreviated tumor subtypes. **d** Correlation analysis of snaR-A gene activity signatures with (1) EMT migration markers or (2) cell proliferation markers. *p*-values were calculated using a permutation test, defined as the proportion of permuted test statistics that were equal to or more extreme than the observed value. Z-score reflects comparison to a randomized null expectation for a given gene set. (EMT markers, $p = 0.857$; Proliferation markers, $p < 10^{-6}$). **e** Mean hazard ratios and corresponding 95% confidence interval for all *SNAR-A* genes (Cox proportional hazard regression model). ($n = 328$ samples). **f** High-to-low ranking of Pol III-transcribed gene "types" on the basis of predicted positive vs. negative roles in cancer. Z-score reflects the observed median Wald statistic for a group "type" compared against a randomization-based null distribution. **g, h** Observations and corresponding empirical null distributions for negative (**g**) and positive (**h**) mutational prognostic frequencies in snaR-A-sensitive, U2 snRNP-bound, and nuclear speckle proximal mRNA subpopulations (p-values determined by permutation test, Top-down: $p < 10^{-6}$, $p = 0.035$, $p = 0.585$, $p = 0.996$, $p < 10^{-6}$, $p = 0.018$, $p = 1$, $p = 1$). Source data are provided as a Source Data file.

A-related splicing defects, including both evidence-based and model-based (prediction) subgroups (Fig. 5g, h). In contrast, these patterns are not evident for U2-bound or speckle-proximal mRNA subpopulations, suggesting snaR-A-disrupted mRNAs represent a select subpopulation with notable importance in disease outcomes (Fig. 5g, h). These findings are altogether consistent with a model in which snaR-A-related changes in cell proliferation are achieved in part through perturbation of splicing events enriched for mRNAs linked to cancer progression.

## Discussion

Here, our survey of snaR-A protein interactions and ensuing functional genomic study provides evidence that snaR-A disrupts mRNA processing through interactions with splicing machinery, potentially through U2 snRNP protein SF3B2. This model is supported by in vitro pull-down experiments, signatures of direct snaR-A RNA-protein cross-linking in SF3B2 PAR-CLIP experiments, and reduced intron retention following snaR-A depletion for transcripts with notable U2 snRNP occupancy. In addition, SF3B2 protein levels are uniquely sensitive to snaR-A overexpression, in contrast to other snaR-A-binding proteins, including ILF3 isoforms NF90 and NF110 (Supplementary Fig. 8) or other U2 snRNP proteins tested (Fig. 1i). We also note that direct visualization of snaR-A revealed nuclear foci in proximity to nuclear speckles (Fig. 2b), indicating that snaR-A-driven splicing perturbation likely takes place at or near mRNA processing sites, rather than through indirect mechanisms such as cytoplasmic sequestration of

splicing machinery. Together, these findings give rise to a model in which snaR-A expression drives processing defects in specific mRNA subpopulations through direct interactions with splicing machinery in the nucleus.

Structurally, SF3B2 is one of several proteins that comprise the SF3b subcomplex, a core component of the U2 snRNP with important roles in spliceosome assembly and stability, branch point recognition, and splicing catalysis[44]. Within SF3b, SF3B2 interactions are largely peripheral, such that SF3B2 may primarily contribute to regulatory and stabilizing interactions rather than the actual catalytic process of splicing[45]. The peripheral nature of SF3B2 raises intriguing possibilities related to the precise mechanism of snaR-A-driven splicing perturbation. For example, snaR-A-triggered U2 instability, by modulation of SF3B2 levels or other mechanisms, would conceivably be maximally disruptive for mRNAs with weak splicing characteristics[46]. The confluence of additional regulatory layers, such as recruitment of specific RBPs and ancillary splicing factors, may ultimately produce the added specificity of snaR-A-related splicing defects.

In addition to U2 residency, nuclear-speckle proximity was a surprisingly important predictor of splicing defects that were improved following snaR-A depletion. This result led us to a counterintuitive finding of higher intron retention levels observed across transcripts produced by speckle-proximal genes (Supplementary fig. 9). Although ectopic targeting of genes to nuclear speckles is shown to increase splicing efficiency (consistent with the expectation of enhanced splicing activity at sites enriched for splicing factors), recent work has linked genes with high G/C content and short introns to inefficient splicing and nuclear speckle retention[27,47]. Though its subnuclear localization provides preliminary evidence that snaR-A may perturb splicing either at or near speckles, the convergence of U2-binding, intron length, sequence content, and speckle proximity altogether obscures whether one feature is particularly significant. Even so, whether the subnuclear coalescence of splicing factors itself is simply a consequence of snRNP retention at unspliced events, thereby driving nucleation of otherwise efficient splicing bodies, remains a possibility.

The significance of snaR-A emergence in cancer is made evident by the detectable signatures linking snaR-A gene activity to unfavorable outcomes in patients. Though the present study makes progress on understanding the molecular and cellular mechanisms through which snaR-A likely promotes disease progression, several new questions are raised. For example, whether splicing disruption and downstream consequences on proliferation are primarily driven through a single or select subset of splicing-defects (i.e. mRNA casualties) may be difficult to address. Though specific GO term enrichments potentially implicate snaR-A in perturbing specific factors involved in autophagy, we find that two indicators of autophagy—p62 receptor levels and LC-I-to-LC-II conversion—remain unchanged following snaR-A depletion in our context (Supplementary Fig. 10). Nevertheless, splicing defects and disrupted protein levels were also observed for multiple tumor suppressor genes, including OGFR (Fig. 4f, g) and HIP1R (Huntingtin Interacting Protein 1-related; Supplementary Fig. 11), two factors known to suppress cell proliferation[48,49]. While it is tempting to draw singular mechanistic paths through such factors, we show that the resolution of processing defects following snaR-A depletion leads to increased protein abundance for these and several other factors, including TIMELESS, a circadian clock protein and oncogenic driver of cancer proliferation, migration, and invasion[50]. Other targets include G3BP2, a stress granule protein that we show increases in abundance—coincident with increased stress granule size—following snaR-A depletion, suggesting snaR-A expression may also interfere with cellular stress responses (Supplementary Fig. 11)[51]. These observations together highlight the true biological complexity as well as the wide-ranging and potentially context-dependent effects snaR-A emergence is likely to have. For example, the physiological role of snaR-A in testes

—another highly proliferative context characterized by dramatic alterations in splicing[10]—may otherwise (or additionally) include cytoplasmic activities linked to the ribosome and translation. Nevertheless, our finding that snaR-A interacts with splicing factors suggests that, in cancer, snaR-A production may essentially introduce "sand into the gears" of mRNA processing. We speculate that various protein and macromolecular vulnerabilities to splicing defects may ultimately allow for opportunistic selection of alterations that promote growth- and survival-related pathways.

**Limitations of this study**

This study focuses on the unexpected finding of splicing factors isolated from snaR-A RNA pull-down experiments and the ensuing proteomic analyses. It is important to note that the in vitro nature of our biotin-snaR RNA pull-down experiments may not accurately reflect the subcellular localization, protein stoichiometry, or cellular modification state of endogenous snaR-A ncRNA, and thus alternative methods of RNA-protein interaction mapping may have captured protein interactions that our approach did not identify. We note that our identification of SF3B2 as a primary interaction candidate was further supported by CLIP and PAR-CLIP experiments. Nevertheless, it is likely that snaR-A interacts with other proteins, including (but not restricted to) additional splicing factors unaccounted for in our study.

Extending from the snaR-A-SF3B2 interaction and co-localization patterns, our ensuing experiments and analyses prioritized the investigation of splicing dynamics inferred by global levels of intron retention. Here, ultra-deep RNA-seq analyses interpreted IR as a generalized "breadcrumb" of splicing efficiency rather than a regulated form of alternative splicing. Though we considered other forms of alternative splicing, including alternative 5' and 3' splice site selection, mutually exclusive exons, and exon skipping, these analyses indicate that changes in IR are the dominant and most directionally biased signature in response to snaR-A (Supplementary Fig. 12). Nevertheless, snaR-A-driven perturbation of splicing is likely to have more nuanced consequences, such as alterations in dominant splicing isoforms extending from other forms of alternative splicing. snaR-A-driven defects may also arise in RNA populations that are not polyadenylated and thus were not captured by our deep mRNA-seq analyses.

While our study focuses on specific molecular signatures related to splicing efficiency, snaR-A-driven consequences are unlikely to be strictly splicing-related. For example, snaR-A emergence may also modulate the steady-state mRNA abundance of specific transcripts, such as through biogenesis of a noncanonical snaR-miRNA recently described to downregulate NME1 RNA level—a negative regulator of metastasis[52]. However, NME1 was not significantly affected in either snaR-A depletion or overexpression experiments in our contexts (Supplementary Fig. 14). Moreover, we show that NME1 is not a negative correlate with snaR-A gene activity in primary tumors, as might be expected for a miRNA-mRNA relationship (Supplementary Fig. 14). Broadly speaking, we find that relatively few transcripts are significantly upregulated or downregulated in response to snaR-A depletion or overexpression, suggesting splicing defects and consequences on protein expression (i.e. translation) are more noteworthy (Supplementary Fig. 13). These findings instead suggest that interactions between snaR-A and splicing factors, and the downstream consequences on cell proliferation are the major determinants of snaR-A-related cancer phenotypes.

Finally, our study made use of available ATAC-seq (chromatin accessibility) datasets to predict snaR-A gene activity in human tumors, given the multiple practical benefits of ATAC (e.g. uniformly applied experimental and sequencing approaches, direct indication of gene-level activity rather than steady-state RNA abundance, etc.). The interpretation of this approach is nevertheless limited by the indirect nature of assessing Pol III transcription through gene accessibility, whereas a direct measure of nascent snaR-A might naturally have been

preferred. Even so, RNA-sequencing methods are highly variable in nature and most often do not include small RNAs <200 nt. The current absence of nascent-seq experiments applied across 100 s of human tumors further precludes any possibility of broadly assessing snaR-A expression and clinical outcome signatures. Thus, future large-scale application of small RNA-seq experiments that include uniform isolation of ncRNAs between 100-200 nt will be necessary to directly assess snaR-A emergence in human cancers.

## Methods

### Cell lines and culture conditions

THP-1 monocytes (ATCC) were cultured in RPMI 1640 Medium (Corning), and HEK293T (ATCC) were cultured in complete Dulbecco's Modified Eagle's Medium (DMEM) (Corning), both supplemented with 10% fetal bovine serum (Corning) and 1% penicillin-streptomycin (Gibco). THP-1 cells were maintained at $2-8 \times 10^5$ cells/mL. HEK293T cells were maintained in 10-cm dishes by seeding $1 \times 10^6$ cells with complete medium every 2–3 days. All cells were maintained in a humidified atmosphere at 37 °C with 5% CO2.

### Plasmids and cell transfection

pcDNA3.1 + /C-(K) DYK plasmids expressing eGFP, SF3B2 (Catalog#O-Hu26570D), La/SSB (Catalog#OHu28669D), NF90 (Catalog# OHu26506D), NF110 (Catalog# OHu26447D), U6-promoter-snaR-A plasmids, and U6-promoter-scramble-snaR-A plasmids were obtained from GenScript. HEK293T cells ($1 \times 10^5$ cells/cm$^2$) were seeded into 12-well Cell-Culture Treated Multidishes (Thermo Scientific) and incubated overnight. Plasmids were transfected using Lipofectamine 3000 (Invitrogen) according to the manufacturer's protocol. Cells were collected 48 h post-transfection.

### Small interfering RNA (siRNA) transfection

siRNAs were mixed with OPTI-MEM media and transfected with Lipofectamine RNAiMAX Transfection Reagent (Invitrogen) per manufacturer's instructions. Three siRNAs against snaR-A and scramble siRNA control (IDT) (Supplementary Table 1) were added at a final concentration of 12.5 nM. RNA was extracted 48 h post-transfection.

### UV cross-linking and immunoprecipitation (CLIP)

HEK293T cells (~$5 \times 10^6$ cells per 60 mm dish, 1 dish per sample) were transfected with FLAG-GFP and FLAG-SF3B2 (GenScript) (14 µg DNA per dish) using Lipofectamine 3000 Transfection Kit (Invitrogen) per manufacturer's instructions. Cells were washed once with 1× PBS (Corning) 48 h post-transfection. Cells were then UV cross-linked at 254 nm with an energy of 400 mJoules/cm$^2$. Cells were lysed with 500ul Pierce IP lysis buffer (Thermo Scientific), supplemented with Halt™ Protease Inhibitor Cocktail (Thermo Scientific) and RNase inhibitor (New England Biolabs) for 5 min on ice. The lysates were cleared by centrifugation at 14,000 $g$ for 10 min at 4 °C. To co-immunoprecipitate RNA-bound FLAG-GFP and FLAG-SF3B2, cleared lysates were mixed with anti-FLAG-antibody-conjugated magnetic beads (Thermo Scientific), and were incubated at room temperature with rotation for 1 hr. Beads were washed two times with high salt wash buffer (50 mM Tris-HCl (pH 7.5), 1 M NaCl, 1 mM EDTA, 1% NP-40, 0.1% SDS), one time with IP lysis buffer (50 mM NaCl, 25 mM Tris (pH=7.5), 1 mM EDTA, 1% NP-40, 5% glycerol) and one time with molecular grade water followed by incubation with proteinase K (New England Biolabs) at 55 °C for 30 min with constant shaking. The bound RNA was extracted using mirVana PARIS RNA and Native Protein Purification kit (Invitrogen) following the manufacturer's instructions for total RNA extraction.

### RNA immunoprecipitation (RIP)

HEK 293T cells (~5 × 10$^6$ cells per 60 mm dish, 1 dish per sample) were transfected with FLAG-GFP, FLAG-NF90, FLAG-NF110, or FLAG-SSB (14 µg DNA per dish) using Lipofectamine 3000 Transfection Kit (Invitrogen) per manufacturer's instruction. Cells were harvested 48 h post-transfection and lysed in Pierce IP lysis buffer (Thermo Scientific), supplemented with Halt™ Protease Inhibitor Cocktail (Thermo Scientific) and RNase inhibitor (New England Biolabs) for 5 min on ice. The lysates were cleared by centrifugation at 14,000 $g$ for 10 min at 4 °C. To co-immunoprecipitate RNA-bound FLAG-GFP, FLAG-NF90, FLAG-NF110, and FLAG-SSB, cleared lysates were mixed with anti-FLAG-conjugated magnetic beads (Thermo Scientific), and the binding reaction was incubated for 1 h at room temperature with rotation. The beads were washed three times with IP lysis buffer (50 mM NaCl, 25 mM Tris (pH 7.5), 1 mM EDTA, 1% NP-40, 5% glycerol) and one time with molecular grade water followed by incubation with proteinase K (New England Biolabs) at 55 °C for 30 min. The bound RNA was extracted using mirVana PARIS RNA and Native Protein Purification kit (Invitrogen) following the manufacturer's instructions for total RNA extraction.

### RNA isolation and RT-qPCR

RNA was isolated using E.Z.N.A.® Total RNA Kit I (Omega Bio-tek), with 100% ethanol instead of 70% ethanol for inclusion of small RNA. For small RNA transcripts, reverse transcription of RNA (10-200 ng) was performed using the TaqMan MicroRNA Reverse Transcription Kit (Applied Biosystems) with the target-specific Taqman™ RT primers (Thermo Fisher) (Supplementary Table 2) according to the manufacturer's instructions. For mRNA, reverse transcription of RNA (1 µg) was performed using High-Capacity cDNA Reverse Transcription Kit (Applied Biosystems). Quantitative PCR (qPCR) was performed using TaqMan™ Fast Advanced Master Mix (Thermo Fisher) and predesigned TaqMan™ Gene Expression Assays (Thermo Fisher) for selected genes. Small RNA abundances were normalized against the geometric mean of control RNA Z30 and U19. mRNA abundance are normalized against the geometric mean of control RNA GAPDH and ACTB. The fold change of each target gene relative to controls was calculated using the Comparative CT Method (ΔΔCT Method).

### Immunoblotting

Transfected cells were lysed with cell disruption buffer (Thermo Scientific) for 5 min on ice, followed by centrifugation at 14,000 $g$ for 5 min at 4 °C. The lysate was separated on a 4–20% Mini-PROTEAN® TGX Stain-Free™ Protein Gels (BIO-RAD), transferred to polyvinylidene difluoride membranes (0.2 um) (Invitrogen). Transferred membranes were blocked with 5% blotting-grade blocker (BIO-RAD) in 1×PBST (Cytiva), followed by incubation with the primary antibodies (Supplementary Table 3) at 4 °C overnight. Membranes were washed 3 × 5 min with 1× PBST and incubated with mouse or rabbit secondary antibodies conjugated with horseradish peroxidase (Invitrogen) for 1 h at room temperatures followed by 3 × 5 min wash by 1× PBST. Proteins were visualized using SuperSignal WestFemto (ThermoScientific) with ChemiDoc™ Touch Imaging System (BIO-RAD). Protein abundances were calculated by normalizing to total protein. Fold changes were calculated against the mean of the controls.

### RNA pulldown and mass spectrometry

THP-1 cells (8 × 10$^6$ cell/sample) were lysed in 300 µl CE buffer (10 mM HEPES, 150 mM KCl, 1 mM EDTA, 0.075%(v/v) NP40, 1 mM DTT, 1 mM PMSF), followed by centrifugation at 14,000 $g$ for 10 min at 4 °C to clear the lysate. 400 pmol Biotinylated snaR-A or biotinylated scramble sequence of snaR-A (IDT) (Supplementary Table 4) were pre-heated to 65 °C for 10 min and cooled down to 4 °C to restore proper secondary structures. Pierce Streptavidin Magnetic Beads (80 µl/sample) (Thermo Scientific) were washed twice with an equal volume of 20 mM Tris (pH 7.5) and resuspended in an equal volume of RNA Capture Buffer (20 mM Tris-HCl (pH 7.5), 1 M NaCl, 1 mM EDTA). Biotinylated snaR-A or the scramble sequence was mixed with pre-washed Pierce Streptavidin Magnetic Beads and incubated at room

temperature with rotation for 30 min. The beads were then washed twice with an equal volume of 20 mM Tris-HCl (pH 7.5) and resuspended in THP-1 cell lysate. The beads were incubated at 4 °C for 2 h. After incubation, beads were washed three times with TEKN buffer (10 mM Tris-HCl (pH 8.0), 1 mM EDTA, 250 mM KCl, 0.1% (v/v) NP40) followed by one last wash with molecular grade water (Lonza). The beads were then subjected to RNase digestion in a 25 μl reaction containing 10 mM Tris-HCl (pH 7.5), 1 mM $MgCl_2$, 40 mM NaCl, and 10 μl of A/T1 RNase (Thermo Scientific), for 1 h at 37 °C. The RNA-bound proteins ($n = 2$) were then released in the reaction and subjected to mass-spectrometry using an UltiMate 3000 RSLCnano system (Thermo Scientific) coupled to a Fusion Orbitrap mass spectrometer (Thermo Scientific) at the University of Illinois, Carver proteomics core. Raw MS/MS data were analyzed using MASCOT (MS/MS Ion Search; Matrix Science). Searches were performed against the UniProt human database 20221205 with the following parameters: enzyme specificity set to trypsin, allowing up to 3 missed cleavages; monoisotopic mass values; unrestricted protein mass; peptide mass tolerance of ±10 ppm; and fragment mass tolerance of ±0.3 Da. Quantitation was performed using the Average [MD] method. Statistical significance was defined as $p < 0.05$.

## Proteomics analysis
Peptide intensities were normalized to the total intensity of each individual sample. Proteins enriched for snaR-A binding were identified using the following criteria: the presence of at least one unique peptide in both snaR-A pulldown samples and a $\log_2$ ratio of normalized intensity (snaR-A pulldown) to normalized intensity in both scramble pulldown and beads pulldown > 2.

## HCR™ RNA FISH-IF
To visualize snaR-A localization relative to nuclear speckles in HEK293T, we performed HCR™ RNA-FISH together with immunofluorescence (IF) to detect SON/SC35 as markers for nuclear speckles and SF3B2. In brief, 24 h after plating of the cells, the samples were rinsed once with 1× PBS (Corning) and then fixed in 4% paraformaldehyde. After fixation, we rinsed the samples twice with 1× PBS, then permeabilized them in 70% ethanol overnight at −20 °C. For hybridization, we rinsed the samples twice with 1× PBS and then 1× SSC. Samples were then pre-hybridized in pre-heated hybridization buffer (Molecular Instruments) for 30 min at 37 °C. The pre-hybridization buffer was then removed and hybridization buffer containing RNA FISH probe targeting snaR-A/U6 (4 nM) was added to the samples and the samples were incubated overnight at 37 °C in a humidified incubator. The probes were designed by Molecular Instruments, Inc. Excess probes were removed by rinsing the samples 3× 5 min with 75%, 50%, and 25% wash buffer (Molecular Instruments) diluted with 5× SSCT (5× sodium chloride sodium citrate, 0.1% Tween 20), followed by 3× 5 min wash with 5×, 2×, and 1× SSCT at room temperature. Samples were then pre-amplified in amplification buffer (Molecular Instruments) for 30 min at room temperature. Hairpin h1 and h2 (molecular instruments) were snap-cooled after incubation at 95 °C for 90 s to room temperature for 30 min. Hairpin solutions were prepared by mixing hairpin h1 and h2 in amplification buffer and added to the samples. Samples were incubated in the dark at room temperature overnight. Excess hairpins were removed by 5× 5 min wash with 5×, 2×, 1× SSCT, and PBST at room temperature. Samples were then blocked with blocking buffer (5% BSA, 5% goat serum in 1× PBST) for 1 hr at room temperature. Samples were washed once with 1× PBST for 5 min followed by incubation of primary antibody against SON and SC35 provided by Dr. Andrew Belmont (University of Illinois) at 1:10000 and 1:1000 dilution ratio respectively. SF3B2 antibody (A5875, ABclonal) was used at 1:1000 dilution ratio. A secondary antibody was applied after 3× 5 min PBST wash for 1 h at room temperature. Samples were again washed with 3× 5 min PBST. We added 50 μg/ml DAPI and Wheat Germ Agglutinin (WGA) to stain for nuclei and cell membrane, respectively. We rinsed the samples twice with 1× PBS, added Vectashield Antifade solution (VectorLabs) and proceeded with imaging on a ZEISS LSM 900 confocal. For each sample, we acquired z-stacks at 0.5 μm intervals using a ×60 oil objective.

## Co-localization analysis
Co-localization analysis of snaR-A relative to nuclear speckle markers (SON, SC35), SF3B2, U6 snRNA is done using JACoP (Just Another Co-localization Plugin) in Fiji v 2.16.0[53,54]. Shortly, nucleus regions are selected, and Costes' randomization is performed to generate the null distribution of Pearson's coefficients by shuffling the pixels. Pearson's coefficient of non-randomized channels is then compared to the null distribution. An aggregate null distribution is the combined analysis of at least 30 nuclei. The mean of Pearson coefficients of the 30 nuclei is calculated as the observation.

## Intron-Exon Junction PCR
Total RNA after scramble and snaR-A knockdown was extracted using E.Z.N.A.® Total RNA Kit I (Omega Bio-tek) according to the manufacturer's instructions. Genomic DNA was removed using DNase (Thermo Scientific). RNA (1 μg) was reverse transcribed using High-Capacity cDNA Reverse Transcription Kit (Thermo Fisher), and PCR was performed using GoTaq® Green Master Mix (Promega). The splicing efficiency is monitored using RT-PCR with primers designed to target the upstream and downstream exons (IDT) (Supplementary table 5). Amplified products were separated on a 1% agarose/TAE gel with SYBR™ Safe DNA Gel staining (Invitrogen) and visualized on a Bio-Rad ChemiDoc Imager. The percent spliced-in (PSI) is quantified as $intensity_{retained}/(intensity_{retained} + intensity_{spliced})$ using ImageLab.

## RNA sequencing and analysis
RNA purified from si-scramble, si-snaR-A#1, #2, #3 treated HEK293T cells using E.Z.N.A.® Total RNA Kit I (Omega Bio-tek) were subjected to library preparation after rRNA depletion and poly(A)+ RNA enrichment using the Kapa Hyper Stranded mRNA library kit (Roche) according to the manufacturer's instructions. All high-throughput sequencing libraries were sequenced for 121 cycles using NovaSeq 6000 at the University of Illinois, Roy J.Carver Biotechnology Center. Poly(A)+ RNA reads were first trimmed using TrimGalore v 0.6.5[55] and then mapped to the transcriptome using quasi-mapper Salmon v 1.5.2[56] to generate read counts. Differentially expressed RNA was determined using edgeR v 3.40.2[57].

## eCLIP/iCLIP-seq analysis
eCLIP-seq FASTQ files were retrieved from the ENCODE web portal (https://www.encodeproject.org/). Reads were first trimmed using TrimGalore and then mapped to the genome (GRCh38) using Bowtie2 v2.5.4[58]. PCR duplicates from ENCODE eCLIP-seq were removed using UMI-Tools v1.0.1[59]. Read counts were extracted over a comprehensive RNAcentral database annotation of noncoding RNA using BEDTools v2.28.0[60–63]. The extracted read counts were then quantile normalized. The fold-enrichment of each RNAcentral entry was calculated against the maximum between corresponding control eCLIP-seq or the average of all control eCLIP-seq datasets.

## Gene "ON"/ "OFF" analysis
Primary solid tumor ATAC-seq alignment bam files were retrieved from the Genomic Data Commons Data Portal (https://portal.gdc.cancer.gov). All ATAC-seq samples (328) were equivalently downscaled to 250 million total reads to ensure the baseline strength for determining statistical significance. Signals were extracted over a comprehensive RNAcentral database annotation of noncoding RNAs and the expected signals were computed as the local background regions including 1 kb (λ1k), 10 kb (λ10k), 100 kb (λ100k), and the

whole genome (λgenome) for each RNAcentral gene coordinate entry. The λ value is thereafter determined as the maximum λ value of the distance set. The probability density function (PDF) utilized is given by $P(X = x) = (e^{-\lambda} * \lambda^x) / x!$. If k represents the observed signal for the specific gene, the *p*-value tied to a specific gene enrichment is calculated using $P(X > k)$. *P*-values were adjusted globally using the Benjamini-Hochberg method. Genes shorter than 1000 bp with a median p-value less than 0.05 across different cancer contexts were considered always "ON." For the remaining genes, those with a median p-value less than 0.05 in at least one cancer subtype were considered "ON/OFF"; otherwise, they were considered "OFF". Cancer type included in the analysis are as listed, TGCT: Testicular Germ Cell Tumors; COAD: Colon adenocarcinoma; HNSC: Head and Neck squamous cell carcinoma; ESCA: Esophageal carcinoma; ACC: Adrenocortical carcinoma; UCEC: Uterine Corpus Endometrial Carcinoma; LUSC: Lung squamous cell carcinoma; STAD: Stomach adenocarcinoma; SKCM: Skin Cutaneous Melanoma; LUAD: Lung adenocarcinoma; BLCA: Bladder Urothelial Carcinoma; CHOL: Cholangiocarcinoma; LIHC: Liver hepatocellular carcinoma; BRCA: Breast invasive carcinoma; KIRC: Kidney renal clear cell carcinoma; LGG: Brain Lower Grade Glioma; KIRP: Kidney renal papillary cell carcinoma; PRAD: Prostate adenocarcinoma; PCPG: Pheochromocytoma and Paraganglioma; CESC: Cervical squamous cell carcinoma and endocervical adenocarcinoma; MESO: Mesothelioma; GBM: Glioblastoma multiforme.

## Survival analysis

Gene-specific hazard ratios (HR) and Wald statistics were determined by applying Cox proportional hazard models to our map of gene "ON/OFF" status, using the 'survival' R package v3.5.3[43]. For each gene, analyses were restricted to cancer subtypes in which both ON and OFF events were present, thereby preventing [1] analysis of genes with constitutively ON or OFF states, and [2] avoiding the influence of tissue-specific characteristics. For example, snaR-A is "ON" in 100% of clinically matched TGCT (Testicular germ cell cancer) tumors (related to expression in testes); and thus, TGCT is not included in the survival analysis for snaR-A. Downstream analyses were restricted to genes <400 bp. For comparison across Pol III-transcribed genes, a median Wald statistic was assigned to cohorts of genes encoding a particular RNA type (e.g. 5S rRNA represents all genes encoding 5S rRNA, as snaR-A represents all snaR-A genes). For each RNA type, a z-score was calculated by comparing the median Wald statistic for a given gene set against an empirically determined null distribution that accounts for the same number of genes.

## RNA secondary structure prediction

snaR-A secondary structure was predicted using the Vienna RNAfold web server with standard settings[64].

## Differential splicing (Intron-centric) analysis

Alternatively, spliced events were identified using rMATs-turbo v4.1.1[65] These events were determined based on reads mapping to splice junctions and are categorized into five types: retained intron (RI), skipped exons (SE), alternative 5′ splice site (A5SS), alternative 3′ splice site (A3SS), and mutually exclusive exons (MXE). To determine the maximum number of overlapping events among three different siRNAs targeting snaR-A, events were ranked by a significance-weighted changes in percent spliced-in (ΔPSI) (-log10(FDR) × ΔPSI). We performed 10,000 permutations to calculate the expected number of overlapping events for the three different siRNAs. Permutations were performed separately for inclusion and exclusion events for each type of splicing event. A p-value cutoff of 0.05 was used to determine the number of events to be considered as overlapping. After identifying overlapping events for the three siRNAs targeting snaR-A, events were further filtered using a minimum false discovery rate (FDR) of <0.05 and a maximum absolute PSI value of >0.1 across all samples.

## Transcript-centric analysis

Global intron retention (IR) ratios were determined using IRFinder v1.3.1 by mapping to the human genome GRCh38 and gencode v44 primary assembly annotation[66]. IR events were thereafter filtered on read quality metrics, including a minimal gene expression score (CPM >1), plus minimal read coverage spanning the flanking 3′ and 5′ exons (>10 across all samples, taken individually). Post-filtered IR ratios were then compared between control (scramble) and the distribution of IR ratios for the three independent siRNAs designed against snaR-A. Individual intron IR scores were thereafter defined as the -log$_{10}$(p-value), determined by Q-test on the distribution of IR ratios across control and si-snaR-A experiments. In scenarios of overlapping intron annotation (i.e. a larger parent or smaller "nested" intron), a median IR score was assigned to a singular intron entity. The significance of transcript-wide IR scores was then assessed by comparing the full distribution of intron IR scores to an empirically derived null distribution for a given number of events (introns). This approach was applied using two frameworks, the first [1] accounting only for introns with positive IR scores (-log$_{10}$(p-value) >0), and the second [2] accounting for all introns, including those with 0 scores. Finally, a list of features with significant transcript-wide reduction in IR was determined as any gene with an adjusted permutation *p*-value < 0.05 in both frameworks [1] and [2] (in total, 136 genes).

## SON TSA-seq (Nuclear Speckle distance) analysis

All available SON TSA-seq experiments with normalized signal counts were retrieved from the 4D Nucleome portal (https://www.4dnucleome.org/)[30,67]. In total, 25 experimental datasets derived from H1 hESC, K562, HFFc6, and HCT116 cells were compiled into an average SON TSA-seq score, thereby representing the average positional relationship between a given gene and nuclear speckles in currently tested cell lines.

## Logistic regression model of IR

Transcripts that showed significant reductions in IR levels (see above) were labeled as 'snaR-A-disruption-sensitive' (136 transcripts), while the remaining transcripts were labeled as 'snaR-A-disruption-insensitive' (10,488 transcripts). A penalized logistic regression model, incorporating both L1 and L2 regularization techniques[68], was trained using 277 features, which included transcript length, median intron length, gene G/C content, distance to nuclear speckles (SON TSA-seq score), and RNA-binding protein (RBP) occupancy data from the ENCODE database using R package caret v 6.0.94. To address the class imbalance between 'sensitive' and 'insensitive' transcripts, an oversampling technique was employed. Model training was performed using leave-one-out cross-validation, with bootstrapping to conduct 100 simulations for model generation. The average area under the receiver operating characteristic (ROC) curve (AUC) was calculated to assess the model's predictive performance on unseen data. A reduced model was constructed using selected variables after the previous variable selection step, including gene length, gene G/C content, median intron length, distance to nuclear speckles, SF3B4 occupancy, and SF3A3 occupancy. A mean predicted probability for each transcript for snaR-A sensitivity was calculated from the reduced model by running 100 simulations and taking the mean between across 100 simulations.

Introns that showed significant reductions in IR levels (see above) were labeled as 'snaR-A-disruption-sensitive' (218 introns), while the remaining introns were labeled as 'snaR-A-disruption-insensitive' (8,558 introns). A separate penalized logistic regression model, also incorporating L1 and L2 regularization, was trained using 277 features, which included intron G/C content, G-quadruplex density predicted

using G4Hunter[69], intron length, and RBP occupancy. Model training and evaluation were conducted similarly, using leave-one-out cross-validation with 100 bootstrapped simulations. The average AUC was calculated to measure the model's predictive performance.

### U2 snRNP-residency analysis

500 bp upstream and downstream of every 3' intron-exon junctions of expressed genes are binned into 100 10 bp binds. Reads mapping to each bin for all the junctions are extracted from SF3B4 (HepG2) and SF3A3 (HepG2) eCLIP-seq (ENCODE) using bedtools. Enrichment scores of the U2 occupancy for 5 bins into the introns of the 3' intron-exon junction was determined using a Poisson framework. Briefly, for 5 bins into the introns counting from the 3' junction each bin, the expected signal within 500 bp upstream ($\lambda$up), 500 bp downstream ($\lambda$down), and anaverage across the entire 1000 bp window for all transcript ($\lambda$whole) is computed. The $\lambda$ value is thereafter determined as the maximum lambda value of the distance set. The probability density function (PDF) utilized is given by P ($X = x$) = $(e^{(-\lambda)} * \lambda^x) / x!$. If k represents the observed signal for the bin, the p-value tied to a specific bin enrichment is calculated using P ($X > k$). Finally, a U2 snRNP residency score is computed as the negative logarithm (base 10) of the minimum p-value among the 5 bins.

### Functional enrichment analysis

Gene Ontology enrichment analysis was performed using the R package clusterProfiler v4.6.2[70] with genome-wide annotation for Human org.Hs.eg.db. The gene set was defined as transcripts with significant transcript-wide ($z < -3$) and/or intron-centric (padj. <0.01) scores and predicted snaR-A sensitivity ($n = 298$) (i.e. both observed and predicted snaR-A sensitivity) and compared against a universe of genes with prediction scores ($n = 10,637$). Enrichments were calculated for GO cellular component (CC), Molecular Function (MF), and Biological Process (BP), and filtered for ontologies with duplicate gene groups and/or <= 3 factors. GO enrichment p-values were adjusted using the Benjamini–Hochberg method.

### Cell proliferation and migration essay

Cell migration is determined using the transwell system (REF 354578; Corning, Inc.) with 8.0-µm diameter pores. HEK293T cells or A549 were plated into 12-well plate and snaR-A function knockdown or over-expression were performed following the above method. After 48 h of treatment, a total of $1 \times 10^4$ HEK293T cells or $5 \times 10^4$ A549 cells were seeded into the upper chamber in serum-deprived medium, and 10% FBS-containing medium was seeded into the lower chamber to induce cell migration. After 24 h of incubation, the cells on the upper surface of the filter were removed, and then the residual cells in the transwell chambers were fixed with 4% paraformaldehyde (thermo scientific) at 4°C for 10 min, stained with Giemsa staining (Sigma-Aldrich) at room temperature for 7 min according to the manufacturer's instruction, and photographed under EVOS M5000 (Invitrogen). Ten fields were selected at random under 20× magnification, and cells were counted in each field using Fiji. The average cell count was used for quantification. At least three biological replicates were included.

Cell proliferation is determined using the Click-iT™ Plus EdU Cell Proliferation Kit for Imaging (Invitrogen). HEK293T cells were plated into 8-well cell culture chamber slides (ibidi, Cat.No:80806). Cells were then treated with siRNA targeting snaR-A as described above. Before harvest, cells were labeled with EdU at the final concentration of 20 µM for 1 h. Then cells were fixed, permeabilized and detected according to the manufacturer's instructions. Cells were imaged under ZEISS LSM 900.

### Statistics and Reproducibility

Experiments performed in this manuscript were repeated independently with 2 or more biological replicates, all attempts at replication were successful. For genomic studies, experiments were performed using three biological replicates processed on the same day. No data were excluded from the analyses. Sample allocation was random with respect to treatment condition, control and treatment group experiments were performed in tandem and subsequently analyzed together. All statistical analyses are described in detail in the relevant Methods sections or figure legends.

### Reporting summary

Further information on research design is available in the Nature Portfolio Reporting Summary linked to this article.

## Data availability
The RNA-seq data generated for this study have been deposited in the NCBI Gene Expression Omnibus under accession number GSE271057, GSE286577 and GSE293548, and are publicly available. LC-MS data are available via ProteomeXchange with identifier PXD054429. Source data for the figures and Supplementary Figs. are provided as a Source Data file. Source data are provided with this paper.

## Code availability
Custom code for analyzing U2 snRNP Residency is available at: https://github.com/VanBortleLab/U2snRNP_ResidencyScore or at Zenodo: https://doi.org/10.5281/zenodo.16544706[71].

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

## Acknowledgements

The authors thank Alvaro Hernandez, Chris Wright, Danman Zhang, and staff at the Carver Biotechnology Center for sequencing services, and administrators of the Carl R. Woese Institute for Genomic Biology (UIUC) Biocluster for computational support. We thank Peter Yau, Justine Arrington, Brian Imai, and the staff at the Carver Biotechnology Center for proteomics services. We thank Prof. Andy Belmont, Prof Kannanga-nattu V. Prasanth, Prof Stephanie Ceman, and members of the Van Bortle lab and the Prasanth lab for helpful suggestions. We thank the Belmont lab for generously sharing anti-SON primary antibodies. This work was supported by the National Institutes of Health, National Human Genome Research Institute (NHGRI) grant R00HG010362 and Roy J. Carver Charitable Trust grant 26-6113 to K.V.B., and National Heart, Lung, and Blood Institute (NHLBI) grant R01HL126845 and Chan-Zuckerberg Bio-hub Chicago's Investigator Award to A.K.

## Author contributions

Study design: S.Z., K.V.B. Data collection: S.Z., S.L., L.M., S.C., R.C., K.V.B. Data analysis: S.Z., S.L., L.M., K.V.B., R.K.C. Data interpretation: S.Z., A.K., K.V.B. Writing: S.Z. and K.V.B. with comments from all authors.

## Competing interests

The authors declare no competing interests
