## [Transparent Peer Review file · Nature Communications]

Cancer-associated snaR-A noncoding RNA interacts with core splicing machinery and disrupts processing of mRNA subpopulations

Corresponding Author: Dr Kevin Van Bortle

Version 0:

Reviewer comments:

Reviewer #1

(Remarks to the Author)

Review

Nature Commun. NCOMMS-24-42427

Sihang Zhou et al./ Van Bortle

Cancer-associated snaR-A noncoding RNA interacts with core splicing machinery and disrupts processing of mRNA subpopulations

This work employs genomic and biochemical approaches to study the molecular interactions and function of snaR-A, a hominid-specific non-coding RNA and potential oncogene. By studying snaR-A expression in various tumors and by reducing its expression through siRNA knockdown in a human cell line, this study yields pertinent and exciting findings, but there are substantial issues that need to be addressed. Chief among these are (1) the unclear relationship between snaR-A and cancer; (2) the heavy dependence on 3 overlapping siRNAs; and (3) the co-localization data, which are puzzling.

Summary of findings:

1. snaR-A is grouped into a set of Pol III genes with a variable 'ON/OFF' state of expression in human tumors (lines 83-85).
2. snaR-A has high abundance in testicular germ cell tumors, colon adenocarcinoma, head and neck squamous cell carcinoma and esophageal carcinoma, and is frequently silent in other cancers (lines 89-93).
3. In cancer subtypes with confident ON and OFF snaR-A expression, snaR-A expression was determined to be a negative factor in cancer survival (lines 96-99).
4. snaR-A directly interacts with SF3B2, a supposed regulatory subunit of the U2 snRNP complex (lines 140-144 and Discussion). In contrast to other U2 snRNP proteins, the SF3B2 protein level was reduced by snaR-A overexpression (Fig 2K and L).
5. snaR-A depletion reduces intron retention in specific mRNA subpopulations (line 162), namely those associated with U2 snRNP proteins (lines 165-167).
6. U2 snRNP residency scores at 3' intron junctions correlate with intron retention sensitivities, allowing identification of snaR-A targeted transcripts (lines 186-188), including MCRIP2.
7. Transcripts with high GC content and short introns may be more susceptible to snaR-A regulation (line 199).
8. mRNA subpopulations affected by snaR-A perturbation are over-represented by histone deacetylases, factors involved in autophagy, and the putative tumor suppressor OGFR. These genes are foreseeably connected to cellular growth and survival related pathways that are beneficial to tumorigenesis.

Specific comments:

1. The last line of the Abstract (line 34) should be rewritten to be more circumspect and restricted to snaR-A 'overactivity' or 'expression'. 'Pol III overactivity' is a broad term and is not specifically measured in this study. The 'novel mechanism' advanced is 'hypothesized' or 'proposed' rather than 'established'.

2. Line 71: reference 18, which includes studies on snaR predicted secondary structures should be included after '...highly structured nature'.
3. Line 77 reads 'Gene accessibility studies correlate with the occupancy of RNA pol III and transcription...', but such studies do not necessarily correlate with snaR-A steady state levels. snaR-A levels should be measured in these cells to confirm the assumption.
4. Line 90: 'snaR-A abundance' is not measured in the study, only Pol III accessibility.
5. Line 94: Does 'emergent' mean the ON state of snaR-A?
6. Supplemental Tables 1 and 2 are long and need to be organized to help the reader, perhaps by grouping proteins by function or known associations. Alternatively, they could be dropped.
7. Line 111: One profound functional significance of the snaR-NF90 interaction is that NF90 (and its dsRBMs) is a chaperone for snaR-A, as explored through NF90 knockdown experiments in Ref 18, acting to stabilize an otherwise labile RNA species (Ref 7).
8. Line 149: The 3 siRNAs used to reduce snaR-A expression are nested, targeting the 3' end and having considerable sequence overlap. Therefore, off-target effects are not controlled for. Confirmation requires truly independent siRNAs and/or an add-back rescue experiment. Also, what is the effect of snaR-A overexpression on splicing?
9. Line 156: 'dynamic' should be defined/explained.
10. In contrast to the text (Lines 203-4), snaR-A and SON do not appear to be predominantly co-localized in IF images shown (Figure 4B)? This counter observation may negate the seemingly counterintuitive finding of higher intron retention levels produced by speckle-proximal genes (Line 286)?
11. Barely any snaR-A is visualized in the cytoplasm despite earlier cell fractionation studies showing a significant proportion of snaR-A (albeit seemingly modified) in the cytoplasm and associated with ribosomes (Ref 18 Fig. 6). Do the authors have an explanation for this discrepancy?
12. Line 188: What is the function of MCRIP2?
13. Line 232: 'following' not 'follow'.
14. The evolution of snaR led to the genesis of novel hCGbeta proteins expressed within the testis (Parrott et al 2011, MCB, doi: 10.1128/MCB.00603-10) and this non-coding RNA likely has other functions outside of the nucleus (see point 11). The Discussion could include comment on possible normal cellular function of snaR-A within the testis, its normal physiologic tissue.
15. Fig. 1 legend: cancer abbreviations should be defined.

In sum, the study does not contain direct mechanistic details of tumorigenesis, and tumor-related genes measured do not appear to be affected in a uniform way. However, the major finding – selective mRNA splicing perturbation by snaR-A – is well supported by detailed experimentation and represents a significant advance in understanding potential biological functions of an emerging non-coding RNA oncogene.

Reviewer #2

(Remarks to the Author)

Reviewer #3

(Remarks to the Author)

The manuscript entitled "Cancer-associated snaR-A noncoding RNA interacts with core splicing machinery and disrupts processing of mRNA subpopulations" analyzes a role of short non-coding RNA - snaR-A in splicing and correlation with cancer progression. The authors mapped the expression of snaR-A in various cancers, determined the protein interaction partners of snaR-A and tested an effect of snaR-A on pre-mRNA splicing. They provided evidence that snaR-A interacts with splicing factors, namely LSm proteins and proteins of the SF3B complex, which is part of the U2 snRNP. They also showed that overexpression of snaR-A reduces the expression of SF3B2. Downregulation of snaR-A increases splicing efficiency of a subset of introns with high GC content.

Despite an interesting topic, my overall impression of the manuscript is rather negative. Most of the conclusions are based on correlations. The authors combine bits of experimental data with various correlations from high throughput studies, but a coherent picture of snaR-A function is missing. Just a few examples:

1. Overexpression of snaR-A reduces SF3B2 expression, but splicing defects are induced by downregulation of snaR-A. Does overexpression of snaR-A cause increased expression of SF3B2? What is the molecular mechanism behind the correlation between snaR-A and SF3B2 expression?
2. What is a molecular mechanism behind the downregulation of snaR-A and improved splicing? The authors suggest that sensitive introns are more occupied by U2 snRNPs. Is U2 snRNP occupancy altered by snaR-A knockdown? An additional explanation provided by the authors is that sensitive introns are in close proximity to splicing speckles. Does their localization change after snaR-A knockdown?
3. Overexpression of snaR-A reduces SF3B2 protein levels. Does overexpression of snaR-A affect the splicing efficiency of sensitive introns?

4. The authors identified LSm proteins as snaR-A interactors and used them to prove that snaR-A interacts with splicing factors. LSm proteins interact with the oligoU tail of U6. Does snaR-A also contain an oligoU tail at the 3' end? Or how does the author explain the interaction between LSm proteins and snaR-A?

5. Similarly, how does the author explain the interaction of SF3B with snaR-A? What part of SF3B2 binds snaR-A?

Minor points:

- I don't understand the purpose of Fig. 1a.

- snaR-A is highly structured and RT-qPCR may not be optimal for detection. Alternative method (e.g. Northern blotting) is essential to confirm snaR-A downregulation (Fig. S3) and in pull-down with SF3B2 (Fig. 2).

- Fig. 3e - The author states that "... we find that several U2 and U4/U6 snRNP proteins are positive predictors of snaR sensitivity, including the core U2 snRNP protein SF3B4" I don't see any U4/U6 proteins but several U5 proteins. Typo in "including".

- Fig. 4b. The authors used amplification-based in situ FISH to detect snaR-A in situ. It's a very sensitive method. Does this mean that snaR-A is low abundant and cannot be detected by classical (without amplification step) FISH? What is the average number of snaR-A molecules per cell? There is no negative control for this experiment.

Reviewer #4

(Remarks to the Author)

This manuscript by Zhou and colleagues studies snaR-A, a small coding RNA transcribed by Pol III, and its impact on RNA splicing in the context of cancer metabolism. Using publicly available data and a chromatin accessibility model, they found that snaR-A is (potentially) differentially expressed among different types of cancer and correlates with unfavorable outcomes. Subsequently, they used an unbiased approach to identify RNA-binding proteins interacting with snaR-A and found several splicing factors of the U2 snRNP, such as SF3B2. Next, they showed that reducing the expression of snaR-A with siRNA affects splicing and specifically decreases intron retention, which they attributed to a splicing defect. The affected introns appear to have specific features related to U2 snRNP occupancy, G-C rich sequences, and subcellular localization to nuclear speckles, previously shown as sites of active splicing. The authors propose that snaR-A expression in cancer cells could promote intron retention through destabilization of splicing factors, such as SF3B2, affecting multiple cellular pathways, yet to be defined.

The manuscript is well written and easy to read, except for some technical parts about regression modeling I am unfamiliar with, and the figures are well designed to fit the text. The study includes a novel effort toward understanding the impact of snaR-A on RNA splicing on a molecular level, which I believe has not yet been explored. The described effects on splicing are clear and well characterized, but the main story feels slightly incomplete for publication. Indeed, beyond the description, there is a critical lack of biological consequences at the cellular level. The authors show multiple affected transcripts with differences in intron retention levels and associated changes in protein levels, but there are no further experiments to investigate if any of the pathways in which those proteins are involved are also perturbed. Only one example is shown in a supplementary figure, potentially showing an increased stress response, but more could be expected to link those data to cancer metabolism. For example, is there any proliferation defect when snaR-A is depleted or overexpressed? MTA1 and AUP1 proteins show increased levels when snaR-A is depleted, but is this translated to a change in autophagy efficiency? Having clear evidence that cellular pathways are directly affected by snaR-A manipulation will considerably increase the impact of this study.

Additional comments:

- Another main criticism is the disrupted coherence of the story developed by the authors. Indeed, they showed that (1) high expression of snaR-A correlates with poor outcomes in cancer and (2) overexpression of snaR-A reduces SF3B2 protein levels, but then, they proceed to build their molecular model in a snaR-A depleted context. It would also be good to explore the defects induced by snaR-A dysregulation in the context of overexpression, which would be more biologically relevant to cancer metabolism. Furthermore, it is unclear why the authors quantified protein levels of splicing factors after snaR-A overexpression but not in the context of snaR-A depletion, for which most of the other analyses of the manuscript were done.

- The authors' description of the global effect of snaR-A on splicing is sometimes confusing. For example, most of the data are explored in the context of snaR-A depletion and the authors always used the terms "splicing defects" or "disruption of mRNA processing" to describe a situation where there is less intron retention than in normal cells. It would be good to explain why fewer retained introns indicate less efficient splicing. Additionally, it is surprising to observe many retained introns in their control conditions. Are those specific introns known to be retained in the literature and/or indicative of efficient splicing?

- The authors directly focus our attention on intron retention defects but never mention any other types of splicing defects in the main text. If perturbing snaR-A impacts the U2snRNP, one could expect other defects, such as usage of alternative of 3' splicing sites. They mentioned different splicing events in the method sections, but they are not presented in any figures. The authors should explain why they ignore the other types of splicing defects in the results section and choose to only focus on

intron retention.

- The authors focus only on RNA splicing and present little data on the impact of snaR-A on gene expression. Beyond differential splicing events, are there differentially expressed genes when snaR-A is depleted? If yes, how many correlate with those having different intron retention levels? There is an analysis of gene ontology of the transcripts with different intron retained but one could appreciate the same kind of analysis using genes with differential expression following snaR-A depletion.

- Studying the correlation between the introns affected with snaR-A depletion and U2 snRNP occupancy is probably trivial since the U2 snRNP is part of the main spliceosome responsible for splicing of > 99% of introns (the rest being U12 snRNP-dependent). In fact, we can see in Figures 5 d-g some examples of transcripts with retained introns and high occupancy of the U2 snRNP near almost all 3' splice sites, but not all introns are retained after snaR-A depletion.

- The RNA-seq was generated with rRNA depletion and polyA+ enrichment resulting in the exclusion of many transcripts with shortened or no polyA-tail, which could be misspliced. Indeed, mRNA with splicing defects, especially intron retention, are known to be rapidly targeted by NMD pathways to be degraded and might not even be captured.

- The authors could show a Western blot documenting the snaR-A / SF3B2 interaction, similar to the one for the La protein presented in Figure 2a. The snaR-A / SF3B2 interaction is part of the main model developed by the authors.

Minor comments:

- In the Introduction it is mentioned that "snaR-A-driven perturbations reduce protein abundance for a wide-ranging set of factors". This is confusing since reduced protein levels are only observed for SF3B2 when snaR-A is overexpressed. Other figures show increased protein levels of different factors when snaR-A level is reduced, contrary to what is mentioned in the introduction. This should be corrected.

- In Figure 1 the authors used ATAC-seq datasets to quantify the accessibility of Pol III to SNAR-A genes. Could the authors also try to correlate this with Pol III occupancy based on already published datasets that they used in their recently published article? (DOI:10.1016/j.molcel.2024.09.019). It might not be relevant if the data are coming from different cancer types, but it could be of use in those cancers with high snaR-A expression.

- Figure 1c: would it be possible to indicate the location of snaR-A in the circle?

- Figure 1d is not referenced in the main text

- Figure 2b: it is confusing whether the "splicing factors (aggregate)" is a real point or a legend for other dots present in the plot. Also, it would be informative to add some factors such as SF3B2 in the plot.

- Figure 2c: in the legend, add the number of biological replicates and the statistical test / p-value threshold used for the number of star character displayed

- In the text – lines 142-143 related to figures 2j-i: "We confirm snaR-A-SF3B2 interactions by CLIP-qPCR and further show that overexpression of snaR-A results in subunit-specific depletion of SF3B2, in contrast to U2 snRNP proteins SF3A1, SF3A3, and SF3B4" This should be separated into two sentences as there is no evident link between the interaction itself and the fact that snaR-A could influence the level of the RBP bound to it.

- Figure 5i: in the legend, it is annotated as "(g)" and not "(i)". Also, the multiple red arrows pointing down next to "increased IR" are confusing.

- In the Methods section describing the CLIP and RIP, it says 5 x 10⁶ cells in a 6 mm dish. Probably a typo with a missing "0" for the dish size?

- In the Methods section, there are repeated sentences at the end of the "Differential splicing (Intron-centric) analysis" paragraph mentioning permutations.

There are multiple typos in the text, especially double spaces or words that are cut into two parts with a dash. The authors may want to double-check again. This is probably a result of editing after applying different styles.

Reviewer #5

(Remarks to the Author)

Version 1:

Reviewer comments:

Reviewer #1

(Remarks to the Author)

This MS was co-reviewed with an Early Career Researcher.

Review

Nature Commun. NCOMMS-24-42427A

Sihang Zhou et al./ Van Bortle

Cancer-associated snaR-A noncoding RNA interacts with core splicing machinery and disrupts processing of mRNA subpopulations

The authors have submitted a revised MS together with additional data and an extensive rebuttal document. The revisions are substantial and satisfactorily answer the questions posed in our initial review. The MS is radically reorganized, which greatly improves its cohesion and flow. Our further comments and suggestions are mostly minor. Provided the issues raised by other reviewers are met, the paper can go forward to publication without further review.

Comments and suggestions

L 25 -26: "Ectopic" expression presumably means overexpression rather than expression in an abnormal place. This should be made clear.

L 88: "Most" is questionable: what about tRNA, 7SL, BC200, Alus...

L 143 for example: "Co-localization" is often understood to mean overlapping in space, but snaR-A appears to be adjacent or proximal to the structures (speckles, splicing components) detected with the protein and U6 RNA markers used. Its distribution seldom overlaps with them. Although this is stated at some places in the MS, the non-overlapping nature of the association should be made clearer.

L 200 et seq.: Fig. 3b shows that some amount of transfected FLAG-tagged SF3B2 is made, but it does not demonstrate that SF3B2 expression is increased as implied by "overexpression". Overexpression, if any, should be demonstrated using SF3B2 antibody (as in Fig. 1i).

L 448: It appears that the NME-1 data applies to its RNA level, and not to its splicing efficiency or protein level. This should be clarified.

L 456: What is the significance of "full length" in this sentence?

The Discussion could be expanded to consider:

- (1) the reason for the adjacent but non-overlapping distribution of snaR-A and other components analyzed (see Rebuttal).
- (2) whether the actions of snaR-A could be mediated in whole or in part by regulating the level of SF3B2.

Reviewer #2

(Remarks to the Author)

Reviewer #4

(Remarks to the Author)

I would like to thank the authors for choosing to address all reviewers' comments comprehensively with new experiments and analysis. All remaining concerns are minor and not worth holding up the manuscript acceptance.

Reviewer #5

(Remarks to the Author)

Reviewer #6

(Remarks to the Author)

Zhou et al. provide a substantially revised version of a manuscript, in which they report that ncRNA snaR-A interacts with pre-mRNA splicing factors, in particular SF3B2 and co-localizes with SF3B2 and U6 snRNA in sub-nuclear foci in the vicinity of nuclear speckles. They find that, while snaR-A genes are located proximal to nuclear speckles, down-regulation of snaR-A leads to reduced intron retention (IR) in genes characterized by extended U2 snRNP residency and speckle proximity (resembling the effect of SF3B2 over-production), enhanced levels of the encoded proteins and decreased cell proliferation, while snaR-A over-production leads to opposite effects. They also observe that snaR-A positively correlates with cell proliferation markers in primary tumors and that snaR-A is a negative predictive factors in cancer.

The manuscript is well-written and should be accessible to a broad audience. The work conducted appears technically sound. The results convincingly link snaR-A levels to splicing efficiency and suggest a link to tumor progression via enhanced cell proliferation. While the molecular mechanism whereby snaR-A modulates splicing efficiency remains unclear, deciphering this mechanism would go beyond the scope of the present manuscript. The observations reported here are certainly interesting in their own right.

The revised manuscript includes a number of additional experiments, in particular studies of the effects of snaR-A over-production, results of which confirm and strengthen the authors' model. In my view, the authors adequately addressed all points raised by the reviewers.

Re: Manuscript NCOMMS-24-42427

Cancer-associated snaR-A noncoding RNA interacts with core splicing machinery and disrupts processing of mRNA subpopulations

We are excited to submit a major revision of our manuscript, NCOMMS-24-42427, titled “Cancer-associated snaR-A noncoding RNA interacts with core splicing machinery and disrupts processing of mRNA subpopulations” by Sihang Zhou et al., for your consideration. This revision includes significant follow-up experiments and analyses prompted by reviewer comments and suggestions, which now add additional support to our initial findings, clarify open questions regarding the significance of snaR-A in cancer, and have otherwise helped to improve the clarity of our report. We thank the reviewers for these critical comments which have helped to strengthen the rigor and impact of our study’s findings.

We are excited by the advance this study represents towards understanding the molecular activities and consequences of snaR-A, a Pol III-derived ncRNA that is silent in human tissues yet emerges concomitantly with the expansion of Pol III transcription in cancer. To date, the molecular activities and consequences of snaR-A have remained largely unaddressed, despite prior reports of its effects on cell phenotype, including enhanced proliferation, migration, and invasion.

To briefly re-summarize the major findings of our report, our study:

- (1) Discovers that snaR-A ncRNA interacts with mRNA splicing factors, particularly with U2 snRNP protein SF3B2.

This discovery is supported by:

- (a) Unbiased, discovery-based biochemical RNA pull-down + mass spectrometry
- (b) Additional RNA pull-down and immunoblot comparison to controls (***new results in response to reviewer comments***)
- (c) Orthogonal genomic analysis of SF3B2 PAR-CLIP data
- (d) Biochemical CLIP-qPCR analysis of SF3B2 and comparison of multiple sRNAs
- (e) In situ visualization of snaR-A and statistically significant co-localization with SF3B2 at nuclear splicing bodies in multiple cell types (***new results in response to reviewer comments***)
 - Nuclear localization of snaR-A is further supported by subcellular fractionation small RNA-seq results (***new results in response to reviewer comments***)

- (2) Discovers that snaR-A perturbs mRNA splicing efficiency, as measured by changes in intron retention - the most direct indicator of global splicing events.

This discovery is supported by:

- (a) snaR-A depletion followed by deep RNA-sequencing and splicing analysis, which shows a reduction in intron retention in cells with reduced snaR-A
- (b) snaR-A **overexpression** followed by deep RNA-sequencing and splicing analysis, which shows an increase in intron retention in cells with elevated snaR-A (***new results in response to reviewer comments***)
- (c) SF3B2 **overexpression** followed by deep RNA-sequencing and splicing analysis, which shows a reduction in intron retention in cells expressing snaR-A (***new results in response to reviewer comments***)

- (d) Exon junction PCR analysis against a multitude of differential splicing events, which support our genomic results and confirm snaR-A-related splicing defects
- (e) Immunoblot analysis against a multitude of differentially spliced transcript protein products, which show that splicing events have consequences on overall protein levels.

(3) Discovers that snaR-A primarily promotes cell proliferation phenotypes that are associated with unfavorable outcomes in cancer patients.

This discovery is supported by:

- (a) Longitudinal measurements of proliferation indices (EdU+ labeling), which show significant reductions in proliferation following snaR-A depletion (***new results in response to reviewer comments***)
- (b) Transwell migration assays, which show significant increases in cell motility following snaR-A depletion, suggesting snaR-A is a driver of proliferation, not migration (***new results in response to reviewer comments***).
- (c) Orthogonal analysis of EMT (migration) and cell proliferation markers in human primary tumors, which detect significant correlations between snaR-A emergence and proliferation markers, but not migration markers (***new results in response to reviewer comments***).
- (d) Survival analysis of snaR-A-related signatures in primary tumors, which show that snaR-A emergence is a negative prognostic factor in cancer.
- (e) Related mutation-based prognostic analysis for mRNA subpopulations perturbed by snaR-A, which shows a clear enrichment (for mRNAs-linked via non-synonymous mutations) to cancer outcomes (***new results, final panel***).

As described above, our revision includes several important follow-up experiments and analyses in response to reviewer comments, which altogether support our initial finding and further clarify the implications and significance of snaR-A-related perturbations in cancer. As a result, we are excited to submit a significantly revised manuscript that addresses previous concerns. We believe these additional experiments and analyses have substantially enhanced our study, and we sincerely appreciate the reviewers' and editor's time, attention, and insightful suggestions.

We would also like to note that, in response to issues of clarity noted in our initial submission, we have also sought to improve the organization and writing of our revision. For example, we now introduce the in situ visualization experiments (including several updates in response to reviewer comments) immediately following our biochemical and genomic discovery of snaR-A - U2 interactions, which naturally support the same discovery linking snaR-A to mRNA splicing factors. Similarly, we have moved the broader cancer survey of snaR-A to the final figure, which is now logically integrated with cell phenotype assays performed in response to reviewer comments. We hope that these changes improve the logical flow and clarity of our report, and again thank the reviewers for helping to improve our study.

Below, we respond point-by-point to each of the individual reviewer comments and concerns.

Reviewer 1

By studying snaR-A expression in various tumors and by reducing its expression through siRNA knockdown in a human cell line, this study yields pertinent and exciting findings, but there are substantial issues that need to be addressed (Chief among) are (1) the unclear relationship between snaR-A and cancer (2) the heavy dependence on 3 overlapping siRNAs and (3) the co-localization data, which are puzzling [...] In sum, the study does not contain direct mechanistic details of tumorigenesis, and tumor-related genes measured do not appear to be affected in a uniform way. However, the major finding - selective mRNA splicing perturbation by snaR-A - is well supported by detailed experimentation and represents a significant advance in understanding potential biological functions of an emerging non-coding RNA oncogene.

We thank reviewer 1 for the positive response regarding the pertinence and value of this study's advance in understanding snaR-A activity. This reviewer shares several important, constructive comments that we have carefully considered in our revision. Importantly, we have performed several follow-up experiments and analyses to address the chief issues noted above, including (1) understanding the role of snaR-A in cancer, (2) our previous model dependence relying solely on results from overlapping siRNAs, and (3) challenges with interpreting the snaR-A in situ visualization and co-localization patterns. Here, we respond to these key issues in detail. Point-by-point responses to all Reviewer 1 comments are posted further below.

To address concern (1), we have performed follow-up cell-based assays to determine phenotypes caused by snaR-A depletion in the same context as our snaR-A study (also in response to Reviewer 3, point 1). By tracking EdU+ cells, we show that snaR-A promotes cell proliferation in our context, consistent with evidence previously reported in multiple human cell types (Lee et al. 2016; Van Bortle et al. 2022; Lee et al. 2017; Ameli Mojarad, Ameli Mojarad, and Pourmahdian 2021; Huang et al. 2018; Shi et al. 2019). However, transwell migration assays demonstrate that snaR-A promotes proliferation but antagonizes cell migration (**Figure 5a-b**, respectively).

Figure 5 (new figure; a-b subset). **snaR-A promotes cell proliferation and is a negative prognostic factor in cancer.** (a) Cell proliferation and EdU (5'-ethynyl-2'-deoxyuridine) incorporation assay following snaR-A depletion (red), compared to scramble control (black). Cell count ($p = 0.0132$), EdU+ count ($p = 0.0018$), and EdU+ fraction (proliferation index determined as percentage of EdU+ nuclei visualized with DAPI, $p = 0.0111$, Two-way ANOVA) were measured at 24, 48, and 72 hours post-transfection. (b) Transwell migration assay and ensuing cell count with giemsa stain following snaR-A depletion (red), compared to scramble control (black). ** $p < 0.001$, t Test.

We complement this new finding with a logical follow-up computational analysis that investigates the relationship between snaR-A signatures in various tumors with internal gene expression markers of cell proliferation and migration. Using this framework, we show that snaR-A signatures are positively (and significantly) associated with cell proliferation markers, but negatively (non-significantly) associated with epithelial-mesenchymal transition markers that would indicate migration (**Figure 5d**, next page). These results are altogether consistent with the cell-based assay results and help to clarify the overall significance of snaR-A emergence and ensuing splicing disruption on cancer-related cell phenotypes.

d snaR-A correlation

Figure 5 (new figure; panel d subset). **snaR-A promotes cell proliferation and is a negative prognostic factor in cancer.** (d) Correlation analysis of snaR-A gene activity signatures with (1) EMT migration markers or (2) cell proliferation markers. Z-score reflects comparison to a randomized null expectation for a given gene set.

To address concern (2), because we are limited by the short ~120nt sequence space of snaR-A and targetable sequence capable of knockdown, we now supplement our initial depletion experiments with snaR-A overexpression studies to determine whether high levels of snaR-A produce contrasting patterns of splicing efficiency. We further performed SF3B2 overexpression to understand the significance of SF3B2 levels on splicing outcomes. These new experiments demonstrate that snaR-A overexpression leads to increased intron ratios, whereas SF3B2 overexpression decreases IR, similar in nature to snaR-A depletion (Figure 3d-h). These new findings provide independent support of our initial model derived solely from siRNA experiments, that snaR-A-centered interactions with SF3B2 and U2 snRNP proteins disrupt splicing outcomes.

Figure 3 (new figure, panel d-e;h subset). **snaR-A expression is associated with higher levels of intron retention (IR), consistent with perturbations of mRNA splicing efficiency.** (d-e) Distributions of transcript-level delta IR z-scores in HEK293T or A549 cells following snaR-A (d) or SF3B2 overexpression (e). (h) Distribution of transcript-level delta IR z-scores in HEK293T cells following snaR-A depletion using siRNA #1, #2, or #3.

To address concern (3), we have performed significant follow up experiments and statistical analysis of in situ snaR-A visualization to better understand the nuclear patterns of snaR-A localization. First, we supplement our initial snaR-A SON (speckle) visualization with new experiments using primary antibodies to SC35, a second marker of nuclear speckles that independently confirms snaR-A subnuclear foci surround these splicing bodies (Figure 2d). Second, we additionally co-stain for both nuclear speckles and SF3B2, the U2 snRNP protein enriched in snaR-A pull-down mass-spec and reciprocally enriched for snaR-A in

PAR-CLIP and CLIP-qPCR experiments. We show that SF3B2 similarly surrounds nuclear speckles, and also localizes to numerous subnuclear foci independent of speckles, as one might expect (**Figure 2d**). These patterns are consistent with previous reports for other spliceosomal snRNP proteins, such as SF3B1 (Girard et al. 2012). Third, we now perform an independent HCR-FISH experiment against the U6 snRNA, and demonstrate that the spliceosomal snRNA similarly surrounds the nuclear speckle, suggesting snaR-A patterns are consistent with core, canonical spliceosomal snRNAs (**Figure 2e**).

Figure 2 (new figure, panel d-e subset). **snaR-A localizes to subnuclear foci that are associated with splicing bodies (speckles) and related factors.** (d) HCR-RNA-FISH detection of snaR-A, co-stained for SC35 (SRSF2, nuclear speckles), SF3B2, and DAPI. (e) HCR-RNA-FISH detection of snaR-A and U6 snRNAs, co-stained for SON (nuclear speckles).

In addition to these additional co-localization analyses, we now quantify the overlapping signal observed between snaR-A, SF3B2, U6 snRNA, and nuclear speckle markers. Our statistical framework applies Costes' randomization to derive an empirical null expectation for overlapping pixels within individual nuclei. By applying this quantitative framework, we now show that snaR-A significantly overlaps with SON, SC35, SF3B2, and U6 snRNAs in HCR-FISH experiments (**Figure 2c**). We further show that patterns observed in HEK293T are similarly observed in A549, a distinct cell type with snaR-A expression (**Supplemental Figure 3**), confirming that snaR-A subnuclear enrichment and proximity to nuclear speckles occurs in multiple contexts.

Figure 2 (new figure; panel c subset). **snaR-A localizes to subnuclear foci that are associated with splicing bodies (speckles) and related factors.** (c) Quantitative analysis of in situ feature co-localization. Observed correlation distributions are compared against Costes randomization-based null hypotheses. Individual plots correspond to co-localization for snaR-A, SON, SC35, SF3B2, and U6 snRNA presented in panels b, d, and e.

Supplemental Figure 3 (new figure; panel c subset). **snaR-A localizes to subnuclear foci that are associated with splicing bodies (speckles) in A549 cells** (c) HCR-RNA-FISH detection of snaR-A in A549, co-stained for SON (nuclear speckles) and DAPI. (left) Quantitative analysis of in situ feature co-localization. Observed correlation distributions are compared against Costes randomization-based null hypotheses (right) Scale bars, 10 μ m

Our revised manuscript includes these and other important follow-up experiments and analyses to address these concerns, which altogether support our initial finding and improve the quality of our study. Below, we address the specific comments raised by reviewer 1 point-by-point:

1. The last line of the Abstract (line 34) should be rewritten to be more circumspect and restricted to snaR-A 'overactivity' or 'expression'. 'Pol III overactivity' is a broad term and is not specifically measured in this study. The 'novel mechanism' advanced is 'hypothesized' or 'proposed' rather than 'established'.

We have revised our abstract and now avoid the use of 'Pol III overactivity' or 'established' as suggested.

2. Line 71: reference 18, which includes studies on snaR predicted secondary structures should be included after '...highly structured nature'.

We have revised this sentence to include the reference noted (this is now re-organized into the final section of our revision (LINE 281).

3. Line 77 reads 'Gene accessibility studies correlate with the occupancy of RNA pol III and transcription...', but such studies do not necessarily correlate with snaR-A steady state levels. snaR-A levels should be measured in these cells to confirm the assumption.

The unavailability of uniform small RNA-seq data generated across tumors (and tissues) is a major limitation of current studies directed at Pol III dynamics. Even more, variability in small RNA preparation (for example, targeting miRNA populations) often omits RNA species between 100-200 nt (snaR is 120 nt), whereas traditional RNA-seq studies do not recover polyA (-) RNA and most often fail to recover RNA species < 200 nt. These shortcomings, which prevent us from measuring snaR-A directly across the TCGA human primary tumor studies surveyed in our report, motivated our application of a chromatin-based survey. The supporting rationale for this framework is derived from in depth proof-of-principle analyses in individual cell types (such as THP-1, example shown below) that demonstrate strong, positive relationships between gene accessibility, measured by ATAC-seq, with Pol III occupancy (ChIP-seq) and small RNA abundance at tRNA genes:

[FIGURE REDACTED]

Fig. 3 Dynamic domain-level regulation of tRNA gene transcription during macrophage differentiation. a Visualization of chromatin and transcriptional dynamics at an example tDNA locus on chromosome 5. Top: in situ Hi-C contact frequency matrix in THP-1 monocytes. *Asterisk represents long-range loop anchor region presented in Fig. 4c. Middle: mean log₂(fold change) signal tracks for chromatin accessibility (ATAC-seq; blue), H3K27 acetylation (ChIP-seq; green), RNA polymerase III occupancy (ChIP-seq; orange), and nascent RNA (Biotin-capture RNA-seq; red) across two adjacent contact domains and neighboring tDNA clusters. Bottom: gene structure and physical contact domain border locations. log₂(fold change) represents ± 72 h PMA treatment.

Genome Biol. 2017; s13059-017-1310-3

While these observations provide evidence that gene accessibility can predict Pol III ChIP-seq and nascent RNA-seq abundance, differential ATAC-seq also mirrors dynamic Pol III and transcription patterns, such as changes in tRNA gene expression in response to THP-1 monocyte-to-macrophage differentiation:

[FIGURE REDACTED]

Fig. 1 Integrated tDNA expression and chromatin profiling in THP-1 monocytes. a Correlation between tRNA gene expression as measured by biotin-capture of nascent, demethylated tRNAs and by RNA polymerase III occupancy mapping by ChIP-seq (black; Spearman's rank correlation coefficient = 0.74; $p < 10^{-16}$). Integrated tDNA expression profile (red) utilizes the mean normalized count for each tRNA gene. b Example signal track representation of the chromatin accessibility (ATAC-seq, blue), active histone signature H3K27 acetylation (ChIP-seq, green), RNA polymerase III occupancy (ChIP-seq, orange), and nascently transcribed RNA (Biotin-capture, red) at a tDNA cluster located on chromosome 6. RPGC mean normalized reads per genomic content. c Correlation between integrated tDNA expression profile with H3K27ac ChIP-seq levels surrounding tRNA genes (black; Spearman's rank correlation coefficient = 0.55; $p < 10^{-16}$) and with chromatin accessibility at tRNA genes as measured by ATAC-seq (blue; Spearman's rank correlation coefficient = 0.79; $p < 10^{-16}$).

Genome Biol. 2017; s13059-017-1310-3

Indeed, these trends have also been shown to be true for other classes of Pol III-transcribed genes, including the broader relationship between gene accessibility and Pol III ChIP-seq across all canonical classes of Pol III-transcribed genes, as well as the relationship between changes in Pol III binding and nascent small RNA-seq:

(LEFT) Supplementary Figure 1 | Pol III subunit expression, genomic occupancy, and chromatin accessibility. (b) Correlation between chromatin accessibility profile (ATAC-seq) and median Pol III subunit signal enrichment.

(RIGHT) Supplementary Figure 2 | Dynamic Pol III subunit expression, occupancy, and compartmental small RNA abundance in THP-1. (h) Correlation between the median Pol III subunit fold change and median change in RNA abundance in THP-1 cells +/- PMA.

Nature Commun. 2022
10.1038/s41467-022-30323-6.

We believe that the relatively small gene size for Pol III-transcribed genes (~70-350nt) underlies the power of chromatin accessibility to predict gene state (“on” or “off”). However, getting directly to the reviewer’s point, these patterns speak to transcription dynamics and may not inform a true picture regarding the level of steady-state small RNA abundance. While our exploitation of gene accessibility seeks to predict snRNA gene activity (in most cases, a simple binary classifier of likely-to-be-active, or likely-to-be-inactive), we have re-examined previous evidence of gene accessibility linked to steady state small RNA abundance

patterns for dynamic snaR-A abundance in specific cells, in this case, the THP-1 leukemia monocytic cell line (below). Here, significant changes in steady-state snaR-A abundance are mirrored by the robust and significant changes in chromatin (gene) accessibility measured by ATAC-seq:

snaR-A chromatin accessibility and steady state (s.s) RNA abundance during THP-1 differentiation Boxplot of snaR-A chromatin accessibility derived from ATAC-seq and snaR-A steady state RNA level from small RNA-seq for THP-1 monocyte and macrophage. P-value calculated using Wilcox test.

Though these and other observations (see response to reviewer 4, minor comment 2) support the use of gene accessibility to predict snaR-A emergence, a necessary procedure given current resources, we also recognize the inherent limitations of this method. We have therefore described this important point in our “Limitations of this Study” subsection linked to our discussion:

Finally, our study made use of available ATAC-seq (chromatin accessibility) datasets to predict snaR-A gene activity in human tumors, given the multiple practical benefits of ATAC (e.g. uniformly applied experimental and sequencing approaches, direct indication of gene-level activity rather than steady-state RNA abundance, etc.). The interpretation of this approach is nevertheless limited by the indirect nature of assessing Pol III transcription through gene accessibility, whereas a direct measure of nascent snaR-A might naturally have been preferred. Even so, RNA-sequencing methods are highly variable in nature and most often do not include small RNAs < 200 nt. The current absence of nascent-seq experiments applied across 100s of human tumors further precludes any possibility of broadly assessing snaR-A expression and clinical outcome signatures. Thus, future large-scale application of small RNA-seq experiments that include uniform isolation of ncRNAs between 100-200 nt will be necessary to directly assess snaR-A emergence in human cancers.

4. Line 90: 'snaR-A abundance' is not measured in the study, only Pol III accessibility.

This is an important point, related to point 3 above, and we have revised the use of ‘snaR-A abundance’ in any reference to experiments derived from gene accessibility in our revision. Please note that these primary tumor studies are now reorganized into the last figure (previously the first figure), to more effectively complement our new experiments regarding cell phenotype.

5. Line 94: Does 'emergent' mean the ON state of snaR-A?

We indeed intended 'emergent' to denote the ON state of snaR-A. We have revised this statement in our revision to improve the clarity of our meaning, and to more precisely convey that this analysis is derived from gene activity prediction:

Original sentence:

To explore the potential impact of emergent SNAR-A expression on patient outcomes, we applied a Cox proportional hazard regression for SNAR-A gene state across cancer, restricting for cancer types with variable SNAR-A expression..

New sentence:

[...] to explore the potential impact of snaR-A on disease progression, we applied a Cox proportional hazard regression for snaR-A gene state across tumors, restricting for the cancer subtypes with variability in predicted gene on or off status.

6. Supplemental Tables 1 and 2 are long and need to be organized to help the reader, perhaps by grouping proteins by function or known associations. Alternatively, they could be dropped.

We thank the reviewer for this suggestion to improve clarity. Our revision has reduced supplemental table 1, which now only includes proteins that are significantly enriched in snaR-A pull-down mass spec. The full supplemental table will instead be provided as an excel sheet at final submission. Likewise, we have removed supplemental table 2, which will instead be provided as an excel sheet at final submission.

7. Line 111: One profound functional significance of the snaR-NF90 interaction is that NF90 (and its dsRBMs) is a chaperone for snaR-A, as explored through NF90 knockdown experiments in Ref 18, acting to stabilize an otherwise labile RNA species (Ref 7).

This is an important point of clarification that we have now addressed in our revision:

Original sentence:

[...] the origin story of snaR-A includes its discovery as an RNA molecule enriched in biochemical pulldown experiments for NF90, a specific isoform of the RNA-binding protein, ILF3(Parrott and Mathews 2007). However, the functional significance of snaR-NF90 interactions and whether additional, presently uncharacterized snaR-A-binding factors are more centrally involved in its potential biological role remain unknown.

New sentence:

[...] the origin story of snaR-A includes its discovery as an RNA molecule enriched in biochemical pulldown experiments for NF90, a specific isoform of the RNA-binding protein, ILF3(Parrott and Mathews 2007). Though NF90 interactions function to chaperone and thereby stabilize snaR-A^{7,18}, whether additional, presently uncharacterized snaR-A-binding factors are more centrally involved in its potential biological role remain unknown.

8. Line 149: The 3 siRNAs used to reduce snaR-A expression are nested, targeting the 3' end and having considerable sequence overlap. Therefore, off-target effects are not

controlled for. Confirmation requires truly independent siRNAs and/or an add-back rescue experiment. Also, what is the effect of snaR-A overexpression on splicing?

This is an important concern (described in the chief issues section above), which prompted our follow-up snaR-A overexpression experiments and splicing analyses, described in detail in our response to primary concern #2 above.

9. Line 156: 'dynamic' should be defined/explained.

We have modified this statement (dynamic IR levels is removed) to improve clarity of meaning:

Original sentence:

[...] PCR analysis confirms dynamic intron retention levels for several rMATS-defined differential splicing events following snaR-A depletion (Supplemental Figure 4).

New sentence:

We show that many such differential IR events are supported by exon-junction PCR experiments for several transcripts, ...

10. In contrast to the text (Lines 203-4), snaR-A and SON do not appear to be predominantly co-localized in IF images shown (Figure 4B)? This counter observation may negate the seemingly counterintuitive finding of higher intron retention levels produced by speckle-proximal genes (Line 286)?

This is an important concern (described in the chief issues section above), which prompted our follow-up co-staining and statistical analysis of snaR-A overlap with nuclear speckle markers, splicing factor SF3B2, and U6 snRNA. These new findings confirm that snaR-A co-localization with all three markers is significant. However, as the reviewer notes, subnuclear snaR-A foci are not restricted to nuclear speckles. It is important to clarify that SF3B2 and U6 snRNAs are similarly characterized by non-speckle loci throughout the nucleus, as splicing processes are not restricted to nuclear speckles. Instead, nuclear speckles represent uniquely enriched rather than exclusive splicing subcompartments. Whereas SON and SC35 are previously characterized “core” speckle (not spliceosomal) proteins, spliceosomal proteins have been reported by others to surround these core markers, as similarly observed for snaR-A and SF3B2 in our report. One such example (focused on catalytically active phospho-SF3B1) is shared below:

Active spliceosomes localize to the periphery of nuclear speckles and are excluded from dense chromatin. Speckles were visualized with anti-SC35 antibody (red), and active spliceosomes with the anti-pT313-SF3b155 antibody (green). Magnification of the boxed region in c. Z-Stack sections were acquired (optical section of 400–500 nm) and deconvolved. All optical sections from the z-stack are shown from top to bottom. The star indicates the optical section shown in c. Scale bar, 1 μ m.

11. Barely any snaR-A is visualized in the cytoplasm despite earlier cell fractionation

studies showing a significant proportion of snaR-A (albeit seemingly modified) in the cytoplasm and associated with ribosomes (Ref 18 Fig. 6). Do the authors have an explanation for this discrepancy?

We have addressed this question with a follow-up fractionation experiment to better understand our visual imaging of snaR-A subnuclear localization. We specifically performed small RNA purification and sequencing following nuclear and cytoplasmic fractionation, and compared the relative enrichment of each small RNA gene class within these two subcompartment populations. As expected, small nucleolar (sno)RNAs are highly enriched in nuclear fractions, whereas tRNAs are highly enriched in the cytoplasmic fraction (Supplemental Figure 3, below). Using this framework, we find that snaR-A is among the significant, nuclear-enriched population of small RNAs. Moreover, we show that nuclear snaR-A RNA is particularly sensitive to the depletion of RNA chaperone protein La (SSB), suggesting a unique dependence of (likely nascent) nuclear snaR-A on La. While these new findings support and reinforce our visualization of snaR-A within subnuclear foci, it must be noted that snaR-A RNA is also captured in cytoplasmic fractions. We hypothesize that potential modifications of snaR-A (and the limited sequence space) enriched in cytoplasmic fractions may prevent hybridization necessary for HCR RNA-FISH, thus precluding any subcellular visualization of snaR-A in the cytoplasm. Thus, we do not believe or articulate any evidence of a discrepancy between our findings and previous work.

Supplemental Figure 3. Subcellular fractionation small RNA-seq captures differential enrichment of snaR-A within the nucleus. (a) Illustration of workflow of small RNA-seq in HEK293T after SSB(La) knockdown (b) Volcano plot of small RNA nucleus/cytoplasmic enrichment (c) Volcano plot visualization of differentially expressed small RNA genes upon SSB(La) knockdown in the total, nuclear, and cytoplasmic cell fraction. snaR-A is highlighted in purple.

MCRIP2 is a putative stress granule protein according to the Alliance of Genome Resources and a putative regulator of the C-terminal-binding protein (CtBP) transcriptional co-repressor (Weng et al. 2019). We now provide additional information on the putative function of MCRIP2 in our revised manuscript. [LINES 242-247]

[...] We show that many such differential IR events are supported by exon-junction PCR experiments for specific transcripts, including those encoding MTA1 (NuRD regulatory subunit), MCRIP2 (putative stress granule protein), OGFR (opioid growth factor receptor), AUP1 (regulator of lipid metabolism and autophagy), and a host of other factors (Figure 4f-g, Supplemental Figure 6)³⁴⁻³⁷.

13. Line 232: 'following' not 'follow'.

We thank the reviewer for catching this mistake, which has been corrected in the re-writing (and-reorganization) of our revision.

14. The evolution of snaR led to the genesis of novel hCGbeta proteins expressed within the testis (Parrott et al 2011, MCB, doi: 10.1128/MCB.00603-10) and this non-coding RNA likely has other functions outside of the nucleus (see point 11). The Discussion could include comment on possible normal cellular function of snaR-A within the testis, its normal physiologic tissue.

We recognize that the functional significance of snaR-A may be highly context-dependent and appreciate this reviewer's suggestion to include a discussion of this true biological complexity. We now include commentary that includes this discussion and important reference in our revised discussion:

[...] These observations together highlight the true biological complexity as well as the wide-ranging and potentially context-dependent effects snaR-A emergence is likely to have. For example, the physiological role of snaR-A in testes - another highly proliferative context characterized by dramatic alterations in splicing [Parrott 2011] - may otherwise (or additionally) include cytoplasmic activities linked to the ribosome and translation [Parrott 2010]. Nevertheless, our finding that snaR-A interacts with splicing factors suggests that, in cancer, snaR-A production may essentially introduce "sand into the gears" of mRNA processing. We speculate that various protein and macromolecular vulnerabilities to splicing defects may ultimately allow for opportunistic selection of alterations that promote growth- and survival-related pathways.

15. Fig. 1 legend: cancer abbreviations should be defined.

This is an important suggestion to improve the clarity of our manuscript, which previously lacked any reference to the full tumor subtype name. To save the space required to list out all 23 tumor subtypes, we now direct the reader in this Figure legend (now **Figure 5c**) to a full listing by name of tumor subtypes included within the revised methods subsection

Reviewer 2 (#2-3)

The authors mapped the expression of snaR-A in various cancers, determined the protein interaction partners of snaR-A and tested an effect of snaR-A on pre-mRNA splicing. They provided evidence that snaR-A interacts with splicing factors, namely LSm proteins and proteins of the SF3B complex, which is part of the U2 snRNP. They also showed that overexpression of snaR-A reduces the expression of SF3B2. Downregulation of snaR-A increases splicing efficiency of a subset of introns with high GC content [...] The authors combine bits of experimental data with various correlations from high throughput studies, but a coherent picture of snaR-A function is missing.

We thank this reviewer for providing critical feedback on our study. As alluded to in this reviewer summary, our report integrates traditional biochemical, cellular, and functional genomic approaches with large-scale analyses that we would respectfully describe as a major strength of our research program and report. For example, our study uses unbiased biochemical pull-down experiments and mass spectrometry to identify snaR-A protein interactions, which we complement with large-scale analyses of (>200) CLIP-seq experiments. Having then hit unexpectedly on snaR-A-U2 snRNP interactions, which are independently supported by both mass spec and genomic CLIP data, we further confirm this finding with CLIP-qPCR and immunoblot analysis in our context of interest. Though all large-scale (and arguably small-scale) analyses are correlative in nature, this interdisciplinary approach helps to either support or refute models that are built on individual lines of evidence. These findings then motivated the inspection of subcellular snaR-A localization in cells, leading to the discovery of snaR-A enrichment at subnuclear foci in proximity to nuclear speckles. Having discovered (1) that snaR-A interacts with splicing proteins and (2) holds a spatial relationship to nuclear speckles, we then performed snaR-A depletion experiments and examined the effect of snaR-A on splicing. Our analysis, which required ultra-deep mRNA sequencing to achieve splicing-level signatures, demonstrates that snaR-A depletion leads to improvements in splicing efficiency for specific mRNA subpopulations linked to cancer progression. These findings are supported by splicing validation experiments directed at numerous examples, as well as follow-up on protein abundance for these factors. While we believe that these experiments serve to build a coherent and important understanding of snaR-A as a molecular antagonist of splicing (a completely new finding!), we have performed several follow-up experiments in response to the comments below to further strengthen this study's characterization of snaR-A activity and understanding of consequences. We have also sought to significantly improve the writing, organization, and clarity of our manuscript to better convey a coherent thought process and the significance of our findings.

Here, we respond to specific comments:

1. Overexpression of snaR-A reduces SF3B2 expression, but splicing defects are induced by downregulation of snaR-A. Does overexpression of snaR-A cause increased expression of SF3B2? What is the molecular mechanism behind the correlation between snaR-A and SF3B2 expression?

To clarify, our study shows that snaR-A depletion leads to improved splicing (splicing defects are resolved). We have sought to make this more clear in our revision by using this language explicitly to describe the effects of snaR-A depletion, such as

In many cases, splicing improvements (i.e. reduced IR following snaR-A depletion) lead to detectable increases in protein abundance (Figure 4g), suggesting snaR-A-related splicing defects modulate both the production of mature mRNA and downstream protein abundances. Among these changes, the splicing improvements and upregulation

of *OGFR* protein levels may be particularly noteworthy in the context of *snaR-A* and cancer, given that *OGFR* is a well-established inhibitor of cell proliferation and tumorigenesis (Cheng et al. 2009).

Regarding the correlation between *snaR-A* and *SF3B2* expression, we now show that *snaR-A* overexpression does not change the level of *SF3B2* expression, suggesting the effects of *snaR-A* on *SF3B2* protein abundance is independent of changes in gene expression. This additional finding helps clarify that the molecular mechanism between *snaR-A* and *SF3B2* levels most likely occurs through direct effects of *snaR-A* on *SF3B2* protein (many hypotheses arise, but the precise molecular mechanism is beyond the scope of the present study).

Supplemental Figure 11 (new figure; panel b subset). **Differential gene expression analysis following *snaR-A* depletion, overexpression and *SF3B2* overexpression.** (b) Volcano plot visualization of differentially expressed gene upon *snaR-A* overexpression in HEK293T and A549 cells. NME1 and *SF3B2* are highlighted in red. Significance calculated using edgeR two-sided exactTest function, Benjamini–Hochberg corrected p-value.

2. Overexpression of *snaR-A* reduces *SF3B2* protein levels. Does overexpression of *snaR-A* affect the splicing efficiency of sensitive introns?

We have followed up on this question in response here as well as in response to Reviewer 1, point 2. We now supplement our initial depletion experiments with *snaR-A* overexpression studies to determine whether high levels of *snaR-A* produce contrasting patterns of splicing efficiency. We further performed *SF3B2* overexpression to understand the significance of *SF3B2* levels on splicing outcomes. These new experiments demonstrate that *snaR-A* overexpression leads to increased intron ratios, whereas *SF3B2* overexpression decreases IR, similar in nature to *snaR-A* depletion (**Figure 3d-h**). These new findings provide independent support for our initial model derived solely from siRNA experiments, that *snaR-A*-centered interactions with *SF3B2* and U2 snRNP proteins disrupt splicing outcomes.

3. What is a molecular mechanism behind the downregulation of snaR-A and improved splicing? The authors suggest that sensitive introns are more occupied by U2 snRNPs. Is U2 snRNP occupancy altered by snaR-A knockdown? An additional explanation provided by the authors is that sensitive introns are in close proximity to splicing speckles. Does their localization change after snaR-A knockdown?

Extending from our biochemical and cell-based results, our study here explores the relationship between snaR-A knockdown (and now, overexpression) with splicing signatures to understand whether specific mRNA subpopulations are disproportionately disrupted by snaR-A. The logistic regression analysis recovers signatures that appear to be meaningful, including (1) higher levels of U2 snRNP occupancy, and (2) higher levels of nuclear speckle proximity. These findings are naturally congruent with our novel discovery that (1) snaR-A interacts with U2 snRNP proteins, and (2) is enriched at subnuclear foci proximal to nuclear speckles. In other words, this large-scale analysis of splicing dynamics independently identifies patterns that relate to the biochemical and cell-based findings of our study. These patterns give rise to a model in which the newly discovered interplay of snaR-A and splicing factors leads to perturbation of splicing outcomes. While this finding represents high-level mechanistic insight, more precise molecular mechanisms guided by hypotheses like “does snaR-A influence U2 snRNP occupancy” or does “snaR-A dictate mRNA subnuclear position” are natural directions for future research.

4. The authors identified LSm proteins as snaR-A interactors and used them to prove that snaR-A interacts with splicing factors. LSm proteins interact with the oligoU tail of U6. Does snaR-A also contain an oligoU tail at the 3' end? Or how does the author explain the interaction between LSm proteins and snaR-A?

This is an important point raised by the reviewer and, indeed, snaR-A is characterized by a notable oligoU tract at the 3' end of its sequence, similar in nature to that of U6 snRNA:

U6 snRNA

GUGCUCGCUUCGGCAGCACAUUAUCUAAAAUUGGAACGAUACAGAGAAGAUUAGCAUGGCCCCUGCGCAAGGAUGACACGCAAUU
CGUGAAGCGUCCAUAUUUUU

SNAR-A

CCGGAGCCAUUGUGGCUCAGGCCGGUUGCGCCUGCCCUCGGGCCUCACGGAGGCGGGGUUCCAGGGCACGAGUUCGAGGCCAGC
CUGGUCCACAUGGGUCGGA AAAAAGGAUUUUUUUU

We expect that, as for U6 snRNA, interactions between snaR-A and Lsm proteins are likely driven by similarities in the 3' end. We expect that future biochemical studies will provide important insights on the biochemical basis of snaR-A::Lsm interaction, as recently described for the U6 snRNA and Lsm proteins (example: Montemayor et al. 2018).

5. Similarly, how does the author explain the interaction of SF3B with snaR-A? What part of SF3B2 binds snaR-A?

The clearest evidence indicative of the nucleotide-level snaR-A::SF3B2 interaction sites is provided by the SF3B2 PAR-CLIP analysis. PAR-CLIP (Photoactivatable-Ribonucleoside-Enhanced Crosslinking and Immunoprecipitation) produces U-to-C nucleotide conversions at direct UV crosslinking sites. We have updated the corresponding figure panel in our revision to more accurately convey the information provided by PAR-CLIP analysis, including the conversion sequence windows (where U-to-C conversion occurrences are present), as well as the maximum PAR-CLIP sequence enrichment site (**Figure 1g** below, *asterisks

denote single nucleotide with maximum enrichment per conversion window). Nevertheless, the biochemical basis of SF3B2::snaR-A interaction will be more difficult to address. For one, though SF3B2 possesses proline-rich domains found in spliceosomal proteins, it does not possess any conventional RNA-binding domain that would be an obvious target for investigation. However, within the U2 snRNP, SF3B2 has been shown to interact with U2 snRNA at SLIIa and SLI through two regions, amino acids 458–530 and 565–598 (Zhang et al. 2020). We show below that sequence alignment between snaR-A and U2 snRNA occurs specifically within these two important regions (SLIIa and SLI) of U2, suggesting a potentially similar interacting interface. It is worth noting that both of these sequence alignments fall within conversion window 1 and conversion window 2 discovered by PAR-CLIP! However, future in-depth biochemical studies directed at understanding the interface of SF3B2, snaR-A, and other RNA species will be necessary to fully understand the biochemical nature of snaR-A::SF3B2 interactions.

g

Figure 1. snaR-A ncRNA interacts with mRNA splicing machinery, including U2 snRNP protein SF3B2. (panel g subset). **(g)** Visualization of SF3B2 PAR-CLIP read pileup across snaR-A gene clusters, and inspection of sequence conversion windows indicative of direct RNA-protein cross-linking sites. (*asterisk denotes coordinate with maximum enrichment).

	SLI	
RNU2	AUCGCUUCUCGGCCUUUUGGCUAAGAUAAGUGUAGUAUCUGUUCUU-----	47
SNAR	-----CCGGAGCCAUUGUGGCUCAGGCCGGUUG---CGCUGCCUCGGGCCUCACGGA	52
	* *	
	SLIIa	
RNU2	-----AUCAGUUUAUAUCUGAUACGUCCUCAUCCGAGGACAAUAUA	90
SNAR	GGCGGGGUUCCAGGGCACGAGUUCGAGGCCAGCCUGGUCCACA-UGGG-----UCGG	104
	* *	
RNU2	UUAAAUGGAUUUUUGGAGCAGGGAGAUGGAAUAGGAGCUUGCUCGUCCACUCCAGCAU	150
SNAR	AAAAAAGGAUUUUUUUU-----	121
	* *	
RNU2	CGACCUGUAUUGCAGUACCUCCAGGAACGGUGCACCC	188
SNAR	-----	121

snaR-A sequence alignment with U2 snRNA Visualization of snaR-A sequence alignment with U2 snRNA using Clustal Omega. Important secondary structures of U2 snRNA, Stem-loop 1(SLI) and Stem-loop IIa (SLIIa) are highlighted.

Minor points:

- I don't understand the purpose of Fig. 1a.

We have removed the visual illustration in Figure 1a, which was meant to highlight the motivating question behind our study. Instead, we have moved the original panel 1b into Figure 1a with slight modifications that now additionally convey the recent evolution of *snaR-A* and this study's goal to understand its effect. We hope that this reorganization and removal of the original Figure 1a helps improve the clarity of our resubmission.

Figure 1 (panel a subset). **snaR-A ncRNA interacts with mRNA splicing machinery, including U2 snRNP protein SF3B2.** (a) Illustrative overview of established *snaR-A* features and the present study aim of delineating the molecular consequence of *snaR-A* expression in cells.

- *snaR-A* is highly structured and RT-qPCR may not be optimal for detection. Alternative method (e.g. Northern blotting) is essential to confirm *snaR-A* downregulation (Fig. S3) and in pull-down with SF3B2 (Fig. 2).

Our study applies reverse-transcription quantitative PCR (RT-qPCR) analyses using a specialized TaqMan assay that relies on unique stem-loop primers adapted specifically for the quantitative analysis of RNA species < 200 nt. The *snaR-A*-specific reagents used here were previously shown to faithfully capture changes in *snaR-A* abundance observed during THP-1 monocyte-to-macrophage differentiation, in line with accurate sequencing-based approaches directed at nascent RNA, total small RNA, nuclear RNA, and cytoplasmic RNA (Van Bortle et al. 2022). In the present study, these RT-qPCR reagents are additionally supported by our follow-up experiments in which *snaR-A* overexpression is accurately captured by RT-qPCR analysis with the specialized TaqMan reagents (Figure 3a, below). All RIP and CLIP-qPCR analyses, which were themselves alternative methods to examine *snaR-A* enrichment patterns observed in genomic eCLIP and PAR-CLIP sequencing experiments, are quantitative differential comparisons of *snaR-A* enrichment.

Figure 3 (new figure, panel a subset). **snaR-A expression is associated with higher levels of intron retention (IR), consistent with perturbations of mRNA splicing efficiency.** (a) RT-qPCR analysis of *snaR-A* RNA and 5S rRNA levels in HEK293T or A549 cells transfected with *snaR-A* (red) or scramble RNA (black) overexpression constructs.

- Fig. 3e - The author states that "... we find that several U2 and U4/U6 snRNP proteins are positive predictors of snaR sensitivity, including the core U2 snRNP protein SF3B4" I don't see any U4/U6 proteins but several U5 proteins. Typo in "including".

We thank the reviewer for catching this misstatement and typo, which we have now addressed in our revision. While U5 proteins are indeed included among the full list of positive predictors of snaR sensitivity, we have sought to improve the clarity of our revised manuscript by integrating the RBP, sequence, and speckle regression analysis into a single panel that better conveys the totality of our model and highlights the most significant predictive features (new **Figure 4** panels a-c, below). This revision now emphasizes the most significant patterns, including U2 occupancy, speckle proximity, and intron length, which we hope improves the coherence of our study to readers.

Figure 4 (panel a-c subset). **snaR-A depletion improves splicing hallmarks in mRNA subpopulations associated with U2 residency and nuclear speckle proximity.** (a) Illustrative overview of intrinsic and extrinsic features considered for logistic regression analysis (left) and model performance of snaR-A-sensitive classification (right). Regression coefficients for notable features, including U2-binding (SF3B4, SF3A3 eCLIP), speckle proximity (SON TSA-seq), and intron length are highlighted. (b-c) U2 snRNP (b) and SON TSA-seq scores (c) binned by predicted intron retention (IR) improvement following snaR-A depletion.

- Fig. 4b. The authors used amplification-based in situ FISH to detect snaR-A in situ. It's a very sensitive method. Does this mean that snaR-A is low abundant and cannot be detected by classical (without amplification step) FISH? What is the average number of snaR-A molecules per cell? There is no negative control for this experiment.

Due to limitations of the amplification method, we are not able to obtain single molecule resolution of snaR-A. However, previous studies of snaR-A in the same context (HEK293) have quantified the average number of snaR-A molecules per cell at ~ 70,000 copies (compared to ~ 200,000 copies of 7SK) (Parrott and Mathews 2007). In addition to the new HCR-RNA FISH experiments and quantifications, we have added the important negative controls for these experiments, including HCR amplification in the absence of snaR-A specific probes (Amplifier only), as well as IF (Secondary antibody only) controls, now included in Supplemental Figure 3:

Supplemental Figure 3 (panel a,b subset). **snaR-A localizes to subnuclear foci that are associated with splicing bodies (speckles) in A549 cells** (a) Negative control for HCR-RNA-FISH with HCR amplifiers only in HEK293T cells, staining for DAPI (blue) and wheat germ agglutinin (WGA; membrane) (green)

Reviewer 4 (#4-5)

This manuscript by Zhou and colleagues studies snaR-A, a small coding RNA transcribed by Pol III, and its impact on RNA splicing in the context of cancer metabolism. Using publicly available data and a chromatin accessibility model, they found that snaR-A is (potentially) differentially expressed among different types of cancer and correlates with unfavorable outcomes. Subsequently, they used an unbiased approach to identify RNA-binding proteins interacting with snaR-A and found several splicing factors of the U2 snRNP, such as SF3B2. Next, they showed that reducing the expression of snaR-A with siRNA affects splicing and specifically decreases intron retention, which they attributed to a splicing defect. The affected introns appear to have specific features related to U2 snRNP occupancy, G-C rich sequences, and subcellular localization to nuclear speckles, previously shown as sites of active splicing. The authors propose that snaR-A expression in cancer cells could promote intron retention through destabilization of splicing factors, such as SF3B2, affecting multiple cellular pathways, yet to be defined. The manuscript is well written and easy to read, except for some technical parts about regression modeling I am unfamiliar with, and the figures are well designed to fit the text. The study includes a novel effort toward understanding the impact of snaR-A on RNA splicing on a molecular level, which I believe has not yet been explored. The described effects on splicing are clear and well characterized, but the main story feels slightly incomplete for publication. Indeed, beyond the description, there is a critical lack of biological consequences at the cellular level. The authors show multiple affected transcripts with differences in intron retention levels and associated changes in protein levels, but there are no further experiments to investigate if any of the pathways in which those proteins are involved are also perturbed. Only one example is shown in a supplementary figure, potentially showing an increased stress response, but more could be expected to link those data to cancer metabolism. For example, is there any proliferation defect when snaR-A is depleted or overexpressed? MTA1 and AUP1 proteins show increased levels when snaR-A is depleted, but is this translated to a change in autophagy efficiency? Having clear evidence that cellular pathways are directly affected by snaR-A manipulation will considerably increase the impact of this study.

Another main criticism is the disrupted coherence of the story developed by the authors. Indeed, they showed that (1) high expression of snaR-A correlates with poor outcomes in cancer and (2) overexpression of snaR-A reduces SF3B2 protein levels, but then, they proceed to build their molecular model in a snaR-A depleted context. It would also be good to explore the defects induced by snaR-A dysregulation in the context of overexpression, which would be more biologically relevant to cancer metabolism. Furthermore, it is unclear why the authors quantified protein levels of splicing factors after snaR-A overexpression but not in the context of snaR-A depletion, for which most of the other analyses of the manuscript were done.

We thank reviewer 4 for sharing several important and constructive comments that have helped us to improve the quality and impact of our characterization of snaR-A. Major points raised include (1) the lack of investigation in our initial report regarding the impact of snaR-A at the cellular level, (2) examination of effects on cellular pathways, including autophagy, which are overrepresented in mRNA subpopulations linked to snaR-A-related splicing defects, and (3) issues with the overall coherence of our study, such as switching to studying the effects of snaR-A depletion (rather than, or in addition to, snaR-A overexpression). We have now performed several follow-up experiments and analyses to address the deficiencies noted above, including (1) understanding the effect of snaR-A on cellular phenotypes in our context, (2) further investigation of cellular pathways (i.e., autophagy) overrepresented in our splicing results, and (3) the addition of snaR-A overexpression experiments, which help to improve coherence as well as confidence in our findings and model. We have additionally strived to improve the writing, organization, and clarity of our revision that we hope more effectively communicates a coherent story.

Here, we respond to these key issues in detail. Point-by-point responses to individual comments are posted further below.

To address concern (1), we have performed follow-up cell-based assays to determine phenotypes caused by *snaR-A* depletion in the same context as our *snaR-A* study (also in response to Reviewer 1, point 1, and copied here again for clarity). By tracking EdU+ cells, we show that *snaR-A* promotes cell proliferation in our context, consistent with evidence previously reported in multiple human cell types (Ameli Mojarad, Ameli Mojarad, and Pourmahdian 2021; Lee et al. 2017; 2016; Liang et al. 2019; Shi et al. 2019). However, transwell migration assays demonstrate that *snaR-A* promotes proliferation but actually antagonizes cell migration (**Figure 5a-b**, respectively).

Figure 5 (new figure; a-b subset). **snaR-A promotes cell proliferation and is a negative prognostic factor in cancer.** (a) Cell proliferation and EdU (5'-ethynyl-2'-deoxyuridine) incorporation assay following *snaR-A* depletion (red), compared to scramble control (black). Cell count (p = 0.0132), EdU+ count (p = 0.0018), and EdU+ fraction (proliferation index determined as percentage of EdU+ nuclei visualized with DAPI, p = 0.0111, Two-way ANOVA) were measured at 24, 48, and 72 hours post-transfection. (b) Transwell migration assay and ensuing cell count with giemsa stain following *snaR-A* depletion (red), compared to scramble control (black). ** p < 0.001, t Test.

We complement this new finding with a logical follow-up computational analysis that investigates the relationship between *snaR-A* signatures in various tumors with internal gene expression markers of cell proliferation and migration. Using this framework, we show that *snaR-A* signatures are positively (and significantly) associated with cell proliferation markers, but negatively (non-significantly) associated with epithelial-mesenchymal transition markers that would indicate migration (**Figure 5d**, next page). These results are altogether consistent with the cell-based assay results and help to clarify the overall significance of *snaR-A* emergence and ensuing splicing disruption on cancer-related cell phenotypes.

Figure 5 (new figure; panel d subset). **snaR-A promotes cell proliferation and is a negative prognostic factor in cancer.** (d) Correlation analysis of *snaR-A* gene activity signatures with (1) EMT migration markers or (2) cell proliferation markers. Z-score reflects comparison to a randomized null expectation for a given gene set.

To address concern (2), we explored the effects of snaR-A on classical markers of autophagy, including p62 abundance and LC3 I-to-II conversion, neither of which were significantly altered by snaR-A expression (new **Supplemental Figure 8**). p62 would otherwise be expected to increase in abundance in response to disrupted lysosomal degradation, whereas LC3 I-to-II conversion would be expected to decrease in response to limited autophagy flux.

Supplemental Figure 8. Overexpression and knockdown of snaR-A do not affect autophagy marker expression levels. (a,b) Immunoblots (a) and quantification (b) of protein levels of autophagy marker LC3 and p62 following snaR-A knockdown in HEK293T cell. (c-e) Immunoblots (c,d) and quantification (e) of protein levels of autophagy marker LC3 and p62 following overexpression in HEK293T and A549. Protein levels are normalized to total protein level. Biological replicates = 3; two-group comparison analyzed with t-test; multi-group comparison analyzed with ANOVA; * $p \leq 0.05$; ** $p \leq 0.01$; *** $p \leq 0.001$; **** $p \leq 0.0001$.

These results indicate that snaR-A has minimal impact on basal autophagy (at least acutely) in our context of interest, but, importantly, help to clarify what is likely to be a highly complex landscape of mRNA targets and effects. We therefore performed additional follow-up analyses to better understand the significance of these mRNA subpopulations in cancer (**Figure 5**, shown below).

Figure 5 (new Figure, g-h subset). snaR-A promotes cell proliferation and is a negative prognostic factor in cancer. (g-h) Observations and corresponding empirical null distributions for negative (g) and positive (h) mutational prognostic frequencies in snaR-A-sensitive, U2 snRNP-bound, and nuclear speckle proximal mRNA subpopulations (p-values determined by permutation test).

Here (above), we examined whether mRNA populations with splicing defects were overrepresented with respect to cancer mutations (missense, including splice site mutations) linked with disease progression. Our analysis captures significant enrichment of mRNAs with cancer mutations that hold prognostic power (in other words, these mRNAs and their dysregulation are observed and linked to alterations in cancer outcomes!). To our surprise, this overrepresentation is specific to snaR-A sensitive mRNAs, and not otherwise true for all U2-bound mRNAs or for all speckle-proximal mRNA populations (**Figure 5, new panels g-h, above**). These new results suggest that snaR-A-related splicing defects are primarily directed at mRNAs with a unique, cancer-relevant fingerprint. We believe that these follow-up experiments and analyses have helped clarify the significance of snaR-A in cancer, significantly improving the impact of our study.

- The authors' description of the global effect of snaR-A on splicing is sometimes confusing. For example, most of the data are explored in the context of snaR-A depletion and the authors always used the terms "splicing defects" or "disruption of mRNA processing" to describe a situation where there is less intron retention than in normal cells. It would be good to explain why fewer retained introns indicate less efficient splicing. Additionally, it is surprising to observe many retained introns in their control conditions. Are those specific introns known to be retained in the literature and/or indicative of efficient splicing?

In response to the reviewer's point, we would like to clarify that in our previous submission, we proposed that reduced intron retention reflects more efficient splicing (in other words, snaR-A is linked to increased IR, whereas depletion of snaR-A reduced intron retention). To avoid potential confusion, we have removed "disruption of mRNA processing" in our revised manuscript. The definition of splicing efficiency, as the correct removal of introns, is also supported in the literature, (Jia, Mu, and Ackerman 2012). Furthermore, to complement our initial findings showing reduced intron retention upon snaR-A depletion, we performed overexpression experiments and observed the opposite effect: increased intron retention. These results further support the interpretation that snaR-A modulates splicing efficiency in a directionally consistent manner. Regarding the presence of retained introns under control conditions, it is important to note that basal levels of intron retention are commonly observed in mammalian cells, as described by (Braunschweig et al. 2014). To further characterize the snaR-A-sensitive genes, we compared them to Group A genes defined by (Wu et al. 2024). These Group A genes are marked by inefficient intron splicing, proximity to nuclear speckles, high G/C content, and short intron length. Notably, we found a significant overlap between snaR-A-sensitive genes and Group A genes, suggesting that these transcripts may share underlying features contributing to less efficient splicing (**Shown below**).

Overlap between snaR-A sensitive genes and previously reported splicing inefficient genes (Wu et al.). Venn diagram showing the overlap between Group A genes (inefficiently spliced genes defined by Wu et al.) and snaR-A-sensitive genes. Fisher's exact test. $p < 2.2e-16$

- The authors directly focus our attention on intron retention defects but never mention any other types of splicing defects in the main text. If perturbing snaR-A impacts the U2snRNP, one could expect other defects, such as usage of alternative of 3' splicing sites. They mentioned different splicing events in the method sections, but they are not presented in any figures. The authors should explain why they ignore the other types of splicing defects in the results section and choose to only focus on intron retention.

Our decision to focus on intron retention is guided by the fact that intron ratios - the basis of IR - represent global splicing outcomes across each and every intron. In other words, IR scores generated by IRFinder are valuable in providing metrics of spliced vs. unspliced mRNA subpopulations, far beyond more extreme examples of regulation intron retention. Building on this rationale, our transcript-level IR analysis facilitates a higher degree of confidence by integrating multiple intron splicing event signatures and applying a statistical framework to determine the probability of observing IR levels of each transcript. Put simply, this bottom-up analysis of transcript-level signatures provides a robust measure of splicing dynamics that is the most logical primary question for investigation. We recognize that other forms of alternative splicing are of interest, and we have included a new supplemental figure that highlights differential events detected by rMATS (**Supplemental Figure 10, below**). However, relatively few significant events are detected for these other splicing patterns, which are not similarly amenable to transcript-level analyses (for example, Alternative 5' splice site usage may occur only once and small subpopulations of transcripts, whereas all spliced mRNAs have detectable and quantifiable intron ratios for deeper analysis).

Supplemental Figure 10. rMATS analysis of alternative splicing events following snaR-A depletion. Barplots indicate the number of significant events that increase (above 0) or decrease (below 0) following snaR-A depletion and the directionality bias when taking such events into account. A5SS = alternative 5' splice site; A3SS = alternative 3' splice site; MXE = mutually exclusive exon; IR = retained

- The authors focus only on RNA splicing and present little data on the impact of snaR-A on gene expression. Beyond differential splicing events, are there differentially expressed genes when snaR-A is depleted? If yes, how many correlate with those having different intron retention levels? There is an analysis of gene ontology of the transcripts with different intron retained but one could appreciate the same kind of analysis using genes with differential expression following snaR-A depletion.

We have revisited our RNA-seq data to include surveys of differential gene expression in our revision. To the reviewer's question regarding splicing outcomes and changes in mRNA abundance, we can confirm that transcripts with improvements in splicing outcomes (i.e., snaR-A depletion leads to reduced intron ratios) show a detectable increase in gene expression following snaR-A depletion (boxplots, below). It is important to note the significance of this observation, both from a technical and biological perspective. Technically speaking, a gene that goes up in expression benefits from higher read coverage, which lends

additional sensitivity to detecting intron retention and conceivably increases the likelihood of capturing and thereby quantifying a higher intron ratio. However, this is the opposite of our result: mRNAs characterized by splicing improvement also lead to higher mRNA abundance. This observation is consistent with a biological model in which unspliced mRNA subpopulations are directed towards RNA turnover through Nonsense-mediated decay mechanisms.

Increased gene expression for snaR-A sensitive group compared to insensitive group upon snaR-A depletion. Boxplot visualization of changes in gene expression upon snaR-A depletion for snaR-A sensitive and insensitive group.

In our revision, we additionally provide volcano plots of the overall differential gene expression signatures following snaR-A depletion in a new supplemental figure that also includes enriched GO terms detected following each treatment (**Supplemental Figure 11, next page below**). These analyses indicate that splicing, rather than dramatic changes in gene expression, are the major consequence of snaR-A abundance, as relatively few genes are notably altered. We also highlight and comment on the observation that NME1 is not affected by snaR-A overexpression or depletion, as recent work indicates a potential role for snaR-A in modulating NME1 (Stribling et al. 2021).

Supplemental Figure 11. Differential gene expression analysis following snaR-A depletion, overexpression and SF3B2 overexpression. (a-c) Volcano plot visualization of differentially expressed gene upon snaR-A knockdown using three different siRNA (a), snaR-A overexpression in HEK293T and A549 (b), SF3B2 in HEK293T and A549 (c). NME1 and SF3B2 are highlighted in red. Significance calculated using edgeR two-sided exactTest function, Benjamini-Hochberg corrected p-value. (d) Gene ontology (GO) enrichment analysis on transcripts identified in panel a-c.

- Studying the correlation between the introns affected with snaR-A depletion and U2 snRNP occupancy is probably trivial since the U2 snRNP is part of the main spliceosome responsible for splicing of > 99% of introns (the rest being U12 snRNP-dependent). In fact, we can see in Figures 5 d-g some examples of transcripts with retained introns and high occupancy of the U2 snRNP near almost all 3' splice sites, but not all introns are retained after snaR-A depletion.

Despite the fact that U2 is a central player in 3' splice site definition and splicing catalysis, we find that the level of U2-mRNA binding is highly variable across all exon-intron boundaries. U2 is equally critical to

splicing outcomes for all introns, but the differential occupancy or residency of U2 otherwise suggests that you are more likely to capture U2 on specific subpopulations of intron-exon boundaries. Though there are certainly exceptions, our analyses demonstrate a clear and significant relationship between snaR-A sensitivity and these subpopulations characterized by U2 occupancy. Our interpretation of this finding is that U2 is more likely to be “captured” by CLIP at introns with low splicing efficiency (a feature that is well known to relate to transcripts with short introns and higher GC content (Braunschweig et al. 2014; Wu et al. 2024) (which are also captured as significant features in our model), and that these populations are thus most susceptible to snaR-A-related perturbations (i.e., they are the weakest and most sensitive splicing events in the nucleus).

- The RNA-seq was generated with rRNA depletion and polyA+ enrichment resulting in the exclusion of many transcripts with shortened or no polyA-tail, which could be misspliced. Indeed, mRNA with splicing defects, especially intron retention, are known to be rapidly targeted by NMD pathways to be degraded and might not even be captured.

We recognize that our survey of snaR-A related splicing defects is broad but not entirely comprehensive. As noted, RNA subpopulations without polyA tails may also be highly sensitive, and we now comment on this aspect in our revised limitations subsection:

snaR-A-driven defects may also arise in RNA populations that are not polyadenylated and thus were not captured by our deep mRNA-seq analyses.

- The authors could show a Western blot documenting the snaR-A / SF3B2 interaction, similar to the one for the La protein presented in Figure 2a. The snaR-A / SF3B2 interaction is part of the main model developed by the authors.

Following this suggestion, we performed additional RNA pulldown assays to confirm the snaR-A interaction with SF3B2. The new immunoblot analysis for SF3B2 is now shown in Supplemental Figure 2 panel A

Supplemental Figure 2. snaR-A interaction with SF3B2 and enrichment across La (SSB) and ILF3 eCLIP experiments, compared against Pol III-transcribed genes. (panel a subset)

(a) Immunoblot analysis of SF3B2 protein following biotin-snaR-A pull-down, compared to empty beads control, biotin-scramble RNA control, and biotin-miR-snaR RNA.

Minor comments:

- In the Introduction it is mentioned that "snaR-A-driven perturbations reduce protein abundance for a wide-ranging set of factors". This is confusing since reduced protein levels are only observed for SF3B2 when snaR-A is overexpressed. Other figures show increased protein levels of different factors when snaR-A level is reduced, contrary to what is mentioned in the introduction. This should be corrected.

We would like to clarify that our hypothesis is the emergence of snaR-A in cancer could affect the protein abundance of mRNA that are sensitive to snaR-A-mediated splicing disruption. This is supported by our finding that splicing improvements (i.e. reduced IR following snaR-A depletion) lead to detectable increases in protein abundance (Figure 4g, below), suggesting snaR-A-related splicing defects modulate both the production of mature mRNA and downstream protein abundances.

Figure 4 (New figure 4, panel g subset). **snaR-A depletion improves splicing hallmarks in mRNA subpopulations associated with U2 residency and nuclear speckle proximity.** (g) Exon- exon junction PCR analysis for intronic transcript features highlighted in panel f, and the corresponding changes in splicing following snaR-A depletion (PSI = percent spliced in; top); corresponding immunoblot analysis of protein abundance for MTA1, MCRIP2, and OGFR shown at bottom.

Regarding the correlation between snaR-A and SF3B2 expression, we now show that snaR-A overexpression does not change the level of SF3B2 expression, suggesting the effects of snaR-A on SF3B2 protein abundance is independent from changes in gene expression. This additional finding helps clarify that the molecular mechanism between snaR-A and SF3B2 levels most likely occurs through direct effects of snaR-A on SF3B2 protein (many hypotheses arise, but the precise molecular mechanism is beyond the scope of the present study).

Supplemental Figure 11 (new figure; panel b subset). **Differential gene expression analysis following snaR-A depletion, overexpression and SF3B2 overexpression.** (b) Volcano plot visualization of differentially expressed gene upon snaR-A overexpression in HEK293T and A549 cells. NME1 and SF3B2 are highlighted in red. Significance calculated using edgeR two-sided exactTest function, Benjamini-Hochberg corrected p-value.

- In Figure 1 the authors used ATAC-seq datasets to quantify the accessibility of Pol III to SNAR-A genes. Could the authors also try to correlate this with Pol III occupancy based on already published datasets that they used in their recently published article? (DOI:10.1016/j.molcel.2024.09.019). It might not be relevant if the data are coming from different cancer types, but it could be of use in those cancers with high snaR-A expression.

This is a great point and we in fact have a separate manuscript currently in preparation (Lizarazo et al.,) that explores the full breadth of RNA polymerase III activity using chromatin accessibility (i.e., similar to our survey here, of snaR-A), across *all classes of* Pol III transcribed genes. To the point raised by the reviewer, we have taken care to compare the accessibility of Pol III transcribed genes in specific contexts with the occupancy of Pol III. Below, we show a summary of these findings, including (1) the observation that increased Pol III binding determined by ChIP-seq predicts corresponding increases in gene accessibility (panel b), that (2) our statistical framework for defining “gene on” and “gene off” is capable of capturing tissue- and tumor-specific gene signatures (panel a, c), and (3) that follow-up ChIP-seq experiments in relevant cells and tissues confirm accessibility-defined signatures (panel d, POLR3A ChIP-seq in neurons and cardiomyocytes recover examples of brain- and heart-specific gene accessibility, respectively).

[FIGURE REDACTED]

ATAC-measured gene accessibility scales with Pol III transcription signatures and recovers tissue-specific Pol III activity patterns. (Lizarazo et al., in preparation). [REDACTED]

- Figure 1c: would it be possible to indicate the location of snaR-A in the circle?

The location of snaR-A genes in this figure span the variable “on” subgroup (the light to darkish red) subsection of the circle. However, for clarity, we have since removed this panel in our revision (Figures regarding the ON/OFF status of snaR-A in cancer has now been presented in Figure 5c. Along with our follow-up investigation of snaR-A effects on cell proliferation.

- Figure 1d is not referenced in the main text

We have referenced the original Figure 1d (**now Figure 5c**) in our latest submission.

New sentence:

[...]We find that *snaR-A* is most frequently active in Testicular Germ Cell Tumors (TGCT; **Figure 5c**), an observation that is consistent with high *snaR-A* abundance reported in human testes. The *snaR-A* gene is otherwise commonly active across > 50% of Colon adenocarcinoma (COAD, 73%), Head and Neck squamous cell carcinoma (HNSC, 69%), and Esophageal carcinoma (ESCA, 51%), and most frequently silent in Glioblastoma multiforme (GBM, 3%), Mesothelioma (MESO, 6%), and Thyroid carcinoma (THCA, 7%; **Figure 5c**).

- Figure 2b: it is confusing whether the "splicing factors (aggregate)" is a real point or a legend for other dots present in the plot. Also, it would be informative to add some factors such as SF3B2 in the plot.

We have updated Figure 1c (originally Figure 2b) to improve clarity. We have removed the aggregated data point (which was a real point, not a legend) to avoid confusion and, in the revised figure, we now highlight snRNP proteins, including SF3B2, along with nuclear speckle-associated proteins identified in (Dopie et al. 2020).

C

Figure 1. (New figure 1, panel c subset) **snaR-A ncRNA interacts with mRNA splicing machinery, including U2 snRNP protein SF3B2.** (c) Minimum log2(intensity ratio) and total unique peptides counts in snaR-A pull-down mass-spec. ILF3, La, splicing- and speckle-associated proteins are highlighted.

- Figure 2c: in the legend, add the number of biological replicates and the statistical test / p-value threshold used for the number of star character displayed

We have updated the legend for Figure 1 (original Figure 2) in our new submission to clarify the replicates and statistics:

New sentence:

*[...]For RIP (d), biological replicates = 2; for CLIP (h) and immunoblot (i), biological replicates = 3; two-group comparison analyzed with t-test; * $p \leq 0.05$; ** $p \leq 0.01$; *** $p \leq 0.001$; **** $p \leq 0.0001$.*

- In the text - lines 142-143 related to figures 2j-i: "We confirm snaR-A-SF3B2 interactions by CLIP-qPCR and further show that overexpression of snaR-A results in subunit-specific depletion of SF3B2, in contrast to U2 snRNP proteins SF3A1, SF3A3, and SF3B4" This should be separated into two sentences as there is no evident link between the interaction itself and the fact that snaR-A could influence the level of the RBP bound to it.

We have revised this statement into two sentences as suggested to avoid this misunderstanding:

Original sentence:

[...]We confirm snaR-A-SF3B2 interactions by CLIP-qPCR and further show that overexpression of snaR-A results in subunit-specific depletion of SF3B2, in contrast to U2 snRNP proteins SF3A1, SF3A3, and SF3B4.

New sentence:

[...] CLIP-qPCR analysis against flag-tagged SF3B2 confirms snaR-A-SF3B2 interactions in human embryonic kidney 293T (HEK293T) cells, in contrast to U6 and U19 snRNAs, which are not enriched in SF3B2 pull-down experiments (Figure 1h). Moreover, we show that snaR-A overexpression results in subunit-specific depletion of SF3B2, in contrast to U2 snRNP proteins SF3A1, SF3A3, and SF3B4 (Figure 1i).

- Figure 5i: in the legend, it is annotated as "(g)" and not "(i)". Also, the multiple red arrows pointing down next to "increased IR" are confusing.

Original Figure 5i (Model) has been removed from our revision to improve clarity.

- In the Methods section describing the CLIP and RIP, it says 5×10^6 cells in a 6 mm dish. Probably a typo with a missing "0" for the dish size?

We have corrected the size of the dish in our method section:

Corrected sentence:

HEK293T cells ($\sim 5 \times 10^6$ cells per 60 mm dish, 1 dish per sample)...

- In the Methods section, there are repeated sentences at the end of the "Differential splicing (Intron-centric) analysis" paragraph mentioning permutations.

We have corrected these repeated sentences in our new submission.

There are multiple typos in the text, especially double spaces or words that are cut into two parts with a dash. The authors may want to double-check again. This is probably a result of editing after applying different styles.

We apologize for the issues with spacing and typos in our original text brought on by a formatting request, this was our fault. We have reviewed our revision to better ensure these typographical issues were resolved. We thank this reviewer for attention to these mistakes and helping improve the clarity of our revision, in addition to our follow-up experiments.

References

1. Ameli Mojarad, Melika, Mandana Ameli Mojarad, and Alireza Pourmahdian. 2021. "Long Non-Coding RNA snaR Promotes Proliferation in EGFR Wild Type Non-Small Cell Lung Cancer Cells." *International Journal of Molecular and Cellular Medicine* 10 (4): 258–64. <https://doi.org/10.22088/IJMCM.BUMS.10.4.258>.
2. Braunschweig, Ulrich, Nuno L Barbosa-Morais, Qun Pan, Emil N Nachman, Babak Alipanahi, Thomas Gonatopoulos-Pournatzis, Brendan Frey, Manuel Irimia, and Benjamin J Blencowe. 2014. "Widespread Intron Retention in Mammals Functionally Tunes Transcriptomes." *Genome Research* 24 (11): 1774–86. <https://doi.org/10.1101/gr.177790.114>.
3. Cheng, Fan, Patricia J. McLaughlin, Michael F. Verderame, and Ian S. Zagon. 2009. "The OGF–OGFr Axis Utilizes the p16INK4a and p21WAF1/CIP1 Pathways to Restrict Normal Cell Proliferation." *Molecular Biology of the Cell* 20 (1): 319–27. <https://doi.org/10.1091/mbc.E08-07-0681>.
4. Dopie, Joseph, Michael J. Sweredoski, Annie Moradian, and Andrew S. Belmont. 2020. "Tyramide Signal Amplification Mass Spectrometry (TSA-MS) Ratio Identifies Nuclear Speckle Proteins." *The Journal of Cell Biology* 219 (9): e201910207. <https://doi.org/10.1083/jcb.201910207>.
5. Girard, Cyrille, Cindy L. Will, Jianhe Peng, Evgeny M. Makarov, Berthold Kastner, Ira Lemm, Henning Urlaub, Klaus Hartmuth, and Reinhard Lührmann. 2012. "Post-Transcriptional Spliceosomes Are Retained in Nuclear Speckles until Splicing Completion." *Nature Communications* 3 (1): 994. <https://doi.org/10.1038/ncomms1998>.
6. Huang, Yiping, Yingying Hu, Zujian Jin, and Zhaojun Shen. 2018. "LncRNA snaR Upregulates GRB2-Associated Binding Protein 2 and Promotes Proliferation of Ovarian Carcinoma Cells." *Biochemical and Biophysical Research Communications* 503 (3): 2028–32. <https://doi.org/10.1016/j.bbrc.2018.07.152>.
7. Jia, Yichang, John C. Mu, and Susan L. Ackerman. 2012. "Mutation of a U2 snRNA Gene Causes Global Disruption of Alternative Splicing and Neurodegeneration." *Cell* 148 (1): 296–308. <https://doi.org/10.1016/j.cell.2011.11.057>.

8. Lee, Jeeyeon, Jin Hyang Jung, Yee Soo Chae, Ho Yong Park, Wan Wook Kim, Soo Jung Lee, Jae-Hwan Jeong, and Seung Hee Kang. 2016. "Long Noncoding RNA snaR Regulates Proliferation, Migration and Invasion of Triple-Negative Breast Cancer Cells." *Anticancer Research* 36 (12): 6289–95. <https://doi.org/10.21873/anticancerres.11224>.
9. Lee, Jeeyeon, Ho Yong Park, Wan Wook Kim, Soo Jung Lee, Jae-Hwan Jeong, Seung Hee Kang, Jin Hyang Jung, and Yee Soo Chae. 2017. "Biological Function of Long Noncoding RNA snaR in HER2-Positive Breast Cancer Cells." *Tumour Biology: The Journal of the International Society for Oncodevelopmental Biology and Medicine* 39 (6): 1010428317707374. <https://doi.org/10.1177/1010428317707374>.
10. Liang, Kun, Ying Yang, Dingjun Zha, Bo Yue, Jianhua Qiu, and Changming Zhang. 2019. "Overexpression of lncRNA snaR Is Correlated with Progression and Predicts Poor Survival of Laryngeal Squamous Cell Carcinoma." *Journal of Cellular Biochemistry* 120 (5): 8492–98. <https://doi.org/10.1002/jcb.28136>.
11. Montemayor, Eric J., Allison L. Didychuk, Allyson D. Yake, Gurnimrat K. Sidhu, David A. Brow, and Samuel E. Butcher. 2018. "Architecture of the U6 snRNP Reveals Specific Recognition of 3'-End Processed U6 snRNA." *Nature Communications* 9 (1): 1749. <https://doi.org/10.1038/s41467-018-04145-4>.
12. Parrott, Andrew M., and Michael B. Mathews. 2007. "Novel Rapidly Evolving Hominid RNAs Bind Nuclear Factor 90 and Display Tissue-Restricted Distribution." *Nucleic Acids Research* 35 (18): 6249–58. <https://doi.org/10.1093/nar/gkm668>.
13. Shi, Zhitian, Dong Wei, Huamei Wu, Jiayun Ge, Xuefen Lei, Zhitang Guo, Renchao Zou, et al. 2019. "Long Non-Coding RNA snaR Is Involved in the Metastasis of Liver Cancer Possibly through TGF- β 1." *Oncology Letters* 17 (6): 5565–71. <https://doi.org/10.3892/ol.2019.10258>.
14. Stribling, Daniel, Yi Lei, Casey M. Guardia, Lu Li, Christopher J. Fields, Pawel Nowialis, Rene Opavsky, Rolf Renne, and Mingyi Xie. 2021. "A Noncanonical microRNA Derived from the snaR-A Noncoding RNA Targets a Metastasis Inhibitor." *RNA (New York, N.Y.)* 27 (6): 694–709. <https://doi.org/10.1261/rna.078694.121>.
15. Van Bortle, Kevin, David P. Marciano, Qing Liu, Tristan Chou, Andrew M. Lipchik, Sanjay Gollapudi, Benjamin S. Geller, Emma Monte, Rohinton T. Kamakaka, and Michael P. Snyder. 2022. "A Cancer-Associated RNA Polymerase III Identity Drives Robust Transcription and Expression of snaR-A Noncoding RNA." *Nature Communications* 13 (1): 3007. <https://doi.org/10.1038/s41467-022-30323-6>.
16. Weng, Jane S., Takanori Nakamura, Hisashi Moriizumi, Hiroshi Takano, Ryoji Yao, and Mutsuhiro Takekawa. 2019. "MCRIP1 Promotes the Expression of Lung-Surfactant Proteins in Mice by Disrupting CtBP-Mediated Epigenetic Gene Silencing." *Communications Biology* 2 (1): 1–11. <https://doi.org/10.1038/s42003-019-0478-3>.
17. Wu, Jinjun, Yu Xiao, Yunzheng Liu, Li Wen, Chuanyang Jin, Shun Liu, Sneha Paul, Chuan He, Oded Regev, and Jingyi Fei. 2024. "Dynamics of RNA Localization to Nuclear Speckles Are Connected to Splicing Efficiency." *bioRxiv*. <https://doi.org/10.1101/2024.02.29.581881>.
18. Zhang, Zhenwei, Cindy L. Will, Karl Bertram, Olexandr Dybkov, Klaus Hartmuth, Dmitry E. Agafonov, Romina Hofele, et al. 2020. "Molecular Architecture of the Human 17S U2 snRNP." *Nature* 583 (7815): 310–13. <https://doi.org/10.1038/s41586-020-2344-3>.

Re: Manuscript NCOMMS-24-42427

Cancer-associated snaR-A noncoding RNA interacts with core splicing machinery and disrupts processing of mRNA subpopulations

Reviewer 1 & 2

We thank the reviewer for their thoughtful evaluation of our revised manuscript and for acknowledging the improvements in organization, clarity, and data presentation. We appreciate the additional comments and suggestions, which we have carefully considered and incorporated into the final revision where appropriate. Below, we address the specific comments raised by reviewer 1 point-by-point:

1. L 25 -26: "Ectopic" expression presumably means overexpression rather than expression in an abnormal place. This should be made clear.

We indeed intended to convey that snaR-A overexpression leads to increased intron ratios. To avoid ambiguity, we have revised the phrase "ectopic expression" to "snaR-A overexpression" in the manuscript to clearly convey our intended meaning.

2. L 88: "Most" is questionable: what about tRNA, 7SL, BC200, Alus...

We agree with the reviewer's point. We have revised the word "Most" to "Many".

3. L 143 for example: "Co-localization" is often understood to mean overlapping in space, but snaR-A appears to be adjacent or proximal to the structures (speckles, splicing components) detected with the protein and U6 RNA markers used. Its distribution seldom overlaps with them. Although this is stated at some places in the MS, the non-overlapping nature of the association should be made clearer.

To clarify this point, we have revised the relevant text to explicitly state that snaR-A signals are predominantly adjacent to, rather than overlapping with nuclear speckles:

New sentence:

[Line 153] We note that HCR-RNA-FISH detects snaR-A localization *near* subnuclear foci, ...

[Line 158] snaR-A enrichment *adjacent to* nuclear speckles is additionally confirmed by orthogonal co-staining of SC35 (SRSF2), ...

4. L 200 et seq.: Fig. 3b shows that some amount of transfected FLAG-tagged SF3B2 is made, but it does not demonstrate that SF3B2 expression is increased as implied by "overexpression". Overexpression, if any, should be demonstrated using SF3B2 antibody (as in Fig. 1i).

To address this concern, we performed immunostaining using an SF3B2-specific antibody to confirm increased expression levels of transfected SF3B2. The results are now included as Supplemental Figure 5.

Supplemental Figure 5 SF3B2 overexpression in HEK293T (a) Immunoblots and quantification of protein levels of SF3B2 following SF3B2 overexpression.

5. L 448: It appears that the NME-1 data applies to its RNA level, and not to its splicing efficiency or protein level. This should be clarified.

We appreciate the reviewer's observation. To improve clarity, we have revised the text to explicitly state that our findings are based on changes in NME-1 RNA abundance and do not reflect splicing efficiency or protein levels.

New sentence:

[...] such as through biogenesis of a noncanonical snaR-miRNA recently described to downregulate NME1 RNA level - a negative regulator of metastasis.

6. L 456: What is the significance of "full length" in this sentence?

"Full-length" was originally used to emphasize that snaR-A disrupts splicing primarily through the entire RNA, in contrast to the truncated snaR-miRNA. However, we have removed the term for conciseness.

7. The Discussion could be expanded to consider:

(1) the reason for the adjacent but non-overlapping distribution of snaR-A and other components analyzed (see Rebuttal).

While we report that snaR-A localizes adjacent to the nuclear speckle – consistent with observations for core splicing factors – it would be difficult to speculate on the precise structural organization of nuclear splicing bodies, which is beyond our expertise or the focus of our study.

(2) whether the actions of snaR-A could be mediated in whole or in part by regulating the level of SF3B2.

We agree with the reviewer that snaR-A's impact on SF3B2 protein levels may contribute to its functional effects, though it is likely not the sole mechanism. We have accordingly revised the text to reflect this interpretation.

New sentence:

For example, snaR-A-triggered U2 instability, by modulation of SF3B2 levels or other mechanisms, would conceivably be maximally disruptive for mRNAs with weak splicing characteristics.